# SAMBLE: Learning Shape-Specific Sampling Strategies for Point Cloud Shapes with Sparse Attention Map and Adaptive Bin Partitioning

## Abstract

Point cloud sampling plays a pivotal role in facilitating efficient analysis of large-scale point clouds. Recently, learning-to-sample methods have garnered growing interest from the community, particularly for their ability to be jointly trained with downstream tasks. However, previous learning-based sampling methods either lead to unrecognizable sampling patterns by generating a new point cloud or biased sampled results by focusing excessively on shape details. Moreover, they all fail to take the natural point distribution variations over different shapes into consideration and learn a similar sampling strategy for all point clouds. In this paper, we propose a **S**parse **A**ttention **M**ap and **B**in-based **Le**arning method (termed SAMBLE) to learn shape-specific sampling strategies for point cloud shapes, striking a superior balance between the overall shape outline and intricate local details for the sampling process. In particular, we first propose sparse attention map by integrating both local and global information. Based on this, multiple point-wise sampling score computation methods are proposed and explored by leveraging heatmaps as a guiding tool. Subsequently, we introduce a binning strategy that partitions points within each point cloud based on these scores. Finally, additional learnable tokens are introduced during the attention computation phase to acquire sampling weights for each bin, thereby enabling the development of shape-specific sampling strategies for an optimized sampling process. Extensive experiments demonstrate that our method adeptly strikes a refined balance between sampling edge points for local details and preserving uniformity in the global shape, leading to superior performance across common point cloud downstream tasks and even in scenarios involving few-point cloud sampling.

## 1 Introduction

Point cloud sampling is a less explored research area within the realm of this data representation. Traditional random sampling (RS) and farthest point sampling (FPS) remain the most commonly employed methods when sampling is required for point cloud learning and processing. With the advancement of neural networks, several methods have emerged for point cloud sampling in a downstream task-oriented learning framework, including S-Net Dovrat et al. (2019), SampleNet Lang et al. (2020), MOPS-Net Qian et al. (2020), etc. However, these methods essentially generate a new, smaller-sized point cloud instead of directly sampling points from the original input, rendering the techniques akin to black boxes of neural network models with limited interpretability. Consequently, discerning geometric patterns in their qualitative results becomes challenging, as their outcomes closely resemble those obtained through random sampling. More recently, APES Wu et al. (2023a) pioneers the direction of using neural networks to learn point-wise sampling scores, with which it subse-

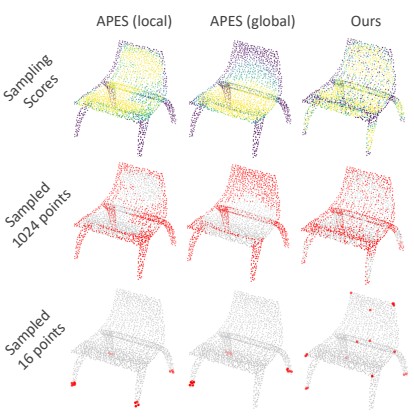

quently samples points whose scores are higher. However, with its score computation design and the Top-M sampling strategy, APES excessively focuses on local details of edge points, resulting in a deficiency in preserving good global uniformity of the input shapes. Consequently, the interpolation operation becomes impractical during the upsampling process, and the sampling quality of few-point sampling is notably subpar. In this paper, we introduce a novel point cloud sampling method that addresses the limitations of prior approaches, aiming to achieve a refined balance between capturing local details and preserving global uniformity.

The concept originates from rethinking the mathematical characteristics of local details within point cloud shapes. Typically, these local details are represented by edge points that define the shape's outline and sharpest features. Is there a point property that can easily distinguish between different categories, such as edge points and non-edge points? The answer is affirmative. In our investigation, we have uncovered an extremely fundamental yet crucial observation: if point $\mathbf{p}_i$ is one of the $k$-nearest neighbors of point $\mathbf{p}_j$, it does not necessarily imply that $\mathbf{p}_j$ is also among the $k$-nearest neighbors of $\mathbf{p}_i$. Consequently, it leads to the conclusion that the *frequency of each point being chosen as a neighbor* exhibits variation across a single point cloud.

We explore and demonstrate the importance of this point property with a simple example as illustrated in Fig. 1. Assume the input point cloud is a simple grid. When selecting 5 neighbors for each point, all three possible cases are given on the left (center point is self-contained as a neighbor). Note that in the triangular and rectangular cases, they each has a "quantum-entangled" twin point pair, in which two points share the possibility of being chosen as the neigh-

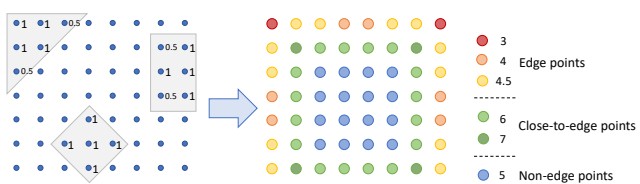

Figure 1: When selecting an equal number of neighbors for each point in the input point cloud, points at different positions are chosen as neighbors with varying frequencies.

bor. While an equal number of neighbors is selected for each point in the input point cloud, points at different positions are chosen as neighbors with varying frequencies, as presented on the right part of Fig. 1. From it, we can observe that in addition to the edge point and non-edge point categories, there is also another noteworthy point category of close-to-edge points. Moreover, within each category, the points can be further grouped into more sub-categories. Overall, this point property effectively captures the local characteristics of a shape, especially for shape outline and sharp details. Building on this point property, we propose a Sparse Attention Map (SAM) and introduce new methods for computing point-wise sampling scores to effectively balance the trade-off between local and global sampling. More details are presented in Sec. 3.2.

On the other hand, after the point-wise sampling scores are computed, previous methods employ a Top-M sampling strategy for all point cloud shapes, which exacerbates the issue of oversampling edge points. We argue that the top-M sampling strategy may not be optimal across all point cloud shapes for downstream tasks. For example, sampling more non-edge points enhances global uniformity, while sampling more close-to-edge points "thickens" the edge, both of which can potentially improve the performance on downstream tasks Wu et al. (2023a). To address this, we introduce a novel bin-based method to explore better sampling strategies shape-specifically by leveraging all point categories. This approach enables the sampling of points with smaller sampling scores, further optimizing the local-global trade-off. As a result, our method dynamically adjusts the sampling strategy for each shape, leading to more tailored and efficient sampling for improved performance.

In this paper, our main contributions can be summarized as follows:

- We propose a sparse attention map that combines the local and global information on the attention map level directly for point cloud sampling. Multiple methods for computing point-wise sampling scores are designed and explored.

- We present a novel method to learn bin boundaries for partitioning points within individual shapes, and tailor shape-specific sampling strategies for them leveraging additional bin tokens.

- The proposed method strikes a better trade-off between sampling local details and preserving global uniformity, leading to better performance both qualitatively and quantitatively.

## 2 RELATED WORK

**Point Cloud Sampling.** Point cloud sampling is a key process in 3D data handling for simplifying high-resolution dense point clouds. Over the past decades, non-learning-based methods Eldar et al. (1997); Moenning & Dodgson (2003); Groh et al. (2018) have predominantly been used for point cloud sampling. While Farthest Point Sampling (FPS) Eldar et al. (1997) is the most widely used one Qi et al. (2017b); Li et al. (2018); Wu et al. (2019); Qian et al. (2022); Zhao et al. (2021), Random Sampling (RS) has also been frequently adopted Zhou & Tuzel (2018); Qi et al. (2020); Groh et al. (2018). More recently, learning-based sampling methods have shown superior performance with task-oriented training. S-Net Dovrat et al. (2019) represents a pioneering work of generating new point coordinates from global representations, while SampleNet Lang et al. (2020) introduces a soft projection operation for better point approximation. Following S-Net, multiple learning-based methods have been proposed Lin et al. (2021); Wang et al. (2021); Nezhadarya et al. (2020); Wang et al. (2023). MOPS-Net Qian et al. (2020) learns a transformation matrix and multiplies it with the original point cloud to generate the sampled one. By employing the attention mechanism to learn point-wise sampling scores, APES Wu et al. (2023a) captures the edge points in the input point clouds with a strong focus.

**Deep Learning on Point Clouds.** In contrast to the voxelization-based methods Maturana & Scherer (2015); Jiang et al. (2018); Le & Duan (2018) and multi-view-based methods Lawin et al. (2017); Boulch et al. (2017); Audebert et al. (2016); Tatarchenko et al. (2018), point-based methods deal directly with point clouds. The pioneer studies of PointNet Qi et al. (2017a) and PointNet++ Qi et al. (2017b) tackle point clouds through point-wise Multi-Layer Perceptrons (MLPs) and max-pooling operations. Subsequently, other research shifts focus towards constructing more efficient building blocks for local feature extraction, such as convolution-based ones Li et al. (2018); Lin et al. (2020a); Zhu et al. (2023); Ahn et al. (2022); Wu et al. (2019); Thomas et al. (2019); Wu et al. (2023b) and graph-based ones Wang et al. (2019); Simonovsky & Komodakis (2017); Chen et al. (2021); Zhang et al. (2021); Xu et al. (2020); Lin et al. (2020b); Liu et al. (2019). More recently, while MLP-based methods like PointNeXt Qian et al. (2022) and PointMetaBase Lin et al. (2023) have rekindled people's interest, the application of attention mechanisms to point cloud analysis has also garnered widespread attention Vaswani et al. (2017); Guo et al. (2021); Zhao et al. (2021); Yu et al. (2022); Engel et al. (2021); Wen et al. (2023); Wu et al. (2024a). For example, PT Zhao et al. (2021); Wu et al. (2022; 2024b) series improves the model performance by introducing subtraction-based attention blocks, and Wu et al. (2024a) performs a large ablation study over attention module designs for point cloud processing.

## 3 METHODOLOGY

A brief pipeline of SAMBLE is illustrated in Fig. 2. It consists of three key steps: constructing a sparse attention map, computing point-wise sampling scores, and learning shape-specific sampling strategies through bin partitioning.

### 3.1 SPARSE ATTENTION MAP

**Local and Global Attention Maps.** Both local and global attention maps are widely used in point cloud analysis. A global attention map is derived from the application of classical self-attention to point features of all points, while a local attention map concentrates on a point-centered area wherein cross-attention is specifically applied to the central point and its neighbors.

Denote $\mathcal{S}_i$ as the set of $k$-nearest neighbors of point $\mathbf{p}_i$, the local attention map for $\mathbf{p}_i$ is defined as

$$\mathbf{m}_i^l = \mathrm{softmax}\left(Q(\mathbf{p}_i)K(\mathbf{p}_{ij} - \mathbf{p}_i)_{j \in \mathcal{S}_i}^\top / \sqrt{d}\right), \tag{1}$$

where $Q$ and $K$ stand for the linear layers applied on the query and key input, and the square root of the feature dimension count $\sqrt{d}$ serves as a scaling factor Vaswani et al. (2017).

For the global attention map which is equivalent to taking all points as the neighbors for each point, it is defined as

$$M^g = \mathrm{softmax}\left(Q(\mathbf{p}_i)K(\mathbf{p}_j)_{i,j \in \mathcal{S}}^\top / \sqrt{d}\right), \tag{2}$$

where $\mathcal{S}$ denotes the set of all input points.

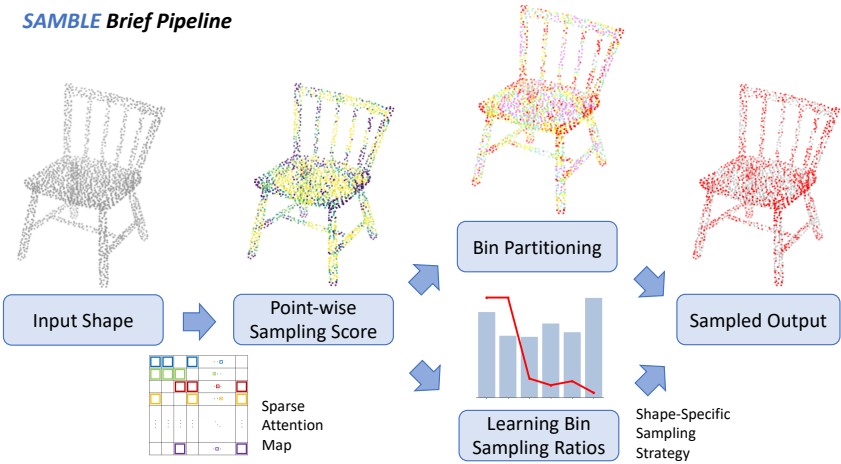

Figure 2: A brief pipeline of our proposed method SAMBLE to learn shape-specific sampling strategies for point cloud shapes.

**Sparse Attention Map.** Instead of using local or global attention maps solely, we propose sparse attention map, which combines the knowledge from both local and global information, to compute point-wise sampling scores. The idea is illustrated in Fig. 3. After obtaining the global attention map with Eq. 2, $KNN$ is employed locally to find $k$ neighbors for each point. In this case, $k$ cells are being selected in each row. However, please notice that if point $\mathbf{p}_j$ is a neighbor to point $\mathbf{p}_i$, it does not mean point $\mathbf{p}_i$ is always also a neighbor to point $\mathbf{p}_j$. This means while for each row $k$ cells are selected, for each column, the number of selected cells varies. The selected cells are then "carved out" to form the sparse attention map, with the values of other non-selected cells being set to 0.

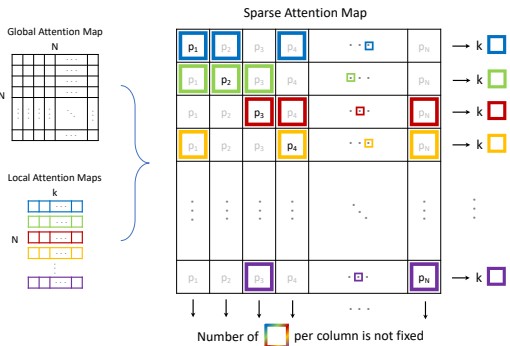

Figure 3: Sparse attention map.

## 3.2 COMPUTING POINT-WISE SAMPLING SCORE

**Indexing Mode.** When sampling points, the points are indexed based on the computed point-wise sampling scores. We call the method of computing point-wise sampling scores from the full/sparse attention map as Indexing Mode. With the original full attention map, following APES, there are two possible indexing modes: (i) row standard deviation; and (ii) column sum. For a global attention map $M^g$ of size $N \times N$, denote $m_{ij}$ as the value of $i$th row and $j$th column in $M^g$. To avoid possible confusion, we use notation $\mathbf{p}_o$ to denote a point only in this subsection. These two indexing modes can be formulated as indexing modes (i) and (ii) in Tab. 1.

With the proposed sparse attention map, there are many other possible indexing modes. As discussed in Sec. 1, to make a better sampling trade-off between sampling edge points and preserving global uniformity, the frequency of each point being chosen as a neighbor, i.e., *the number of selected cells in each column* is the key. We consider the following ones for comparison: (iii) sparse row standard deviation; (iv) sparse row sum; (v) sparse column sum; (vi) sparse column average; and (vii) sparse column square-divided. Again, for a sparse attention map $M^s$ of size $N \times N$, denote $m_{ij}^s$ as the value of $i$th row and $j$th column in $M^s$. For point $\mathbf{p}_o$, we denote the set of indexes of the selected $k$ cells (indexes of KNN neighbors) in $o$th row as $S_o$, and denote the number of selected cells in $o$th column as $n_o$. Details and respective formulas of these indexing modes are listed in Tab. 1.

Table 1: Proposed different indexing modes for computing point-wise sampling scores.

| Indexing Mode | Attention Map | Formula | Remark |
|---|---|---|---|
| (i) Row standard deviation | Full | $a_{\mathbf{P}_o} = f_{\text{std}}(\{m_{oj}|j = 1, 2, \ldots, N\})$ | $f_{\text{std}}$: Computes standard deviation for a set of values |
| (ii) Column sum | Full | $a_{\mathbf{P}_o} = \sum_{i=1}^{N} m_{io}$ | |
| (iii) Row standard deviation | Sparse | $a_{\mathbf{P}_o} = f_{\text{std}}(\{m_{oj}^s|j \in S_o\})$ | $S_o$: Set of indices of selected cells in $o$th row |
| (iv) Row sum | Sparse | $a_{\mathbf{P}_o} = \sum_{j=1}^{N} m_{oj}^s$ | Non-selected cells are all of 0s |
| (v) Column sum | Sparse | $a_{\mathbf{P}_o} = \sum_{i=1}^{N} m_{io}^s$ | |
| (vi) Column average | Sparse | $a_{\mathbf{P}_o} = \sum_{i=1}^{N} m_{io}^s/n_o$ | $n_o$: Number of selected cells in $o$th column |
| (vii) Column square-divided | Sparse | $a_{\mathbf{P}_o} = \sum_{i=1}^{N} m_{io}^s/n_o^2$ | $n_o$: Number of selected cells in $o$th column |

Figure 4: Point sampling score heatmaps under different indexing modes.

**Heatmap.** To analyze the behavior of each indexing mode, we train a separate model for each mode, ensuring that all other settings remain consistent. The sampling score distributions are depicted as heatmaps in Fig. 4, offering additional insights. From these heatmaps, we can see that both row-standard-deviation-based modes (i and iii) concentrate heavily on edge points. However, because they consistently prioritize thin or detailed regions, some areas may be overlooked. In contrast, modes ii and iv show less emphasis on edge points and instead distribute focus across a broader range of points, with a tendency toward other non-edge regions in a biased manner.

More interestingly, the comparison of modes v, vi, and vii, which utilize column-wise information from SAM, reveals distinct sampling preferences and strategies across different point categories. Mode v prioritizes non-edge points, mode vi emphasizes the global shape, and mode vii focuses slightly more on edge points. This is because edge points typically have a smaller number of $n_o$. Despite these differences and unique characteristics, all three modes capture the overall shape more uniformly compared to the former four. In our case, we aim to sample edge points without over-emphasizing them. For instance, when sampling detailed areas like chair legs, we want to capture some edge points without selecting them all, while also ensuring that non-edge points are sampled to preserve better global uniformity. Given this balance, we chose mode vii as the primary indexing mode for most of the experiments in the following sections. The detailed ablation study over different indexing modes is presented in Sec. 4.4.

## 3.3 SAMPLING WITH BINS

After point-wise sampling scores are computed with SAM, points are sampled based on certain rules. The simplest way is to sample points with larger scores, i.e. top-M sampling. In our case, as we aim to enhance the local-global trade-off and leverage all point categories during the sampling process, we suggest employing a bin-based sampling strategy to allow for the sampling of certain close-to-edge points or even non-edge points.

**Bin Partitioning.** The process begins with the processing of the distribution of normalized point-wise sampling scores $a_{\mathbf{P}_i}$ across the shapes within the current batch. Denoting $n_b$ as the number of bins we used for partitioning, $n_b - 1$ boundary values are obtained from this distribution. In each training step, a vector $\boldsymbol{\nu}_c = (\nu_1, \nu_2, \cdots, \nu_{n_b-1})$ that ensures the equitable division of points among

**Algorithm 1** Determining $\boldsymbol{\kappa}$ from $\boldsymbol{\beta}$ and $\boldsymbol{\omega}$

**Require:** number of total points to be selected: $M$, Sampling weights $\boldsymbol{\omega} : [\omega_1, \omega_2, \ldots, \omega_{n_b}]$, number of points in bins $\boldsymbol{\beta} : [\beta_1, \beta_2, ..., \beta_{n_b}]$

1: $\boldsymbol{\kappa} \leftarrow \mathbf{0}$
2: $\mathbf{x} \leftarrow \boldsymbol{\omega} \cdot \boldsymbol{\beta} + \epsilon$
3: $M_r \leftarrow M$
4: **while** $M_r > 0$ **do**
5:     $s \leftarrow \frac{M_r}{\sum x_j}$
6:     **for** $j = 1$ to $n_b$ **do**
7:         $\kappa_j \leftarrow \text{round}(\kappa_j + s x_j)$
8:         **if** $\kappa_j \geq \beta_j$ **then**
9:             $\kappa_j \leftarrow \beta_j$
10:            $x_j \leftarrow 0$
11:         **end if**
12:     **end for**
13:     $M_r \leftarrow M - \sum \kappa_j$
14: **end while**
15: **return** $\boldsymbol{\kappa}$

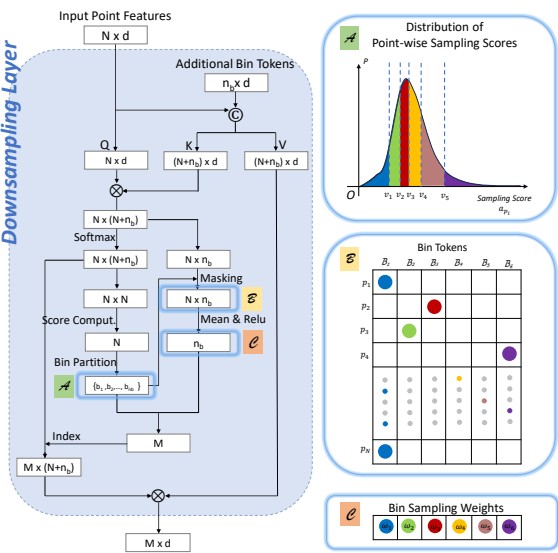

Figure 5: Network structure of our proposed downsampling layer. Block $\mathcal{A}$: Points in each shape are partitioned into $n_b$ bins. Block $\mathcal{B}$: Masking the split-out point-to-token sub-attention map. Block $\mathcal{C}$: Learned bin sampling weights.

all shapes within the current batch is computed based on the point score distribution. Note that even while $\boldsymbol{\nu}_c$ enables an even division across the shapes of the current batch, for each individual shape, points are not evenly partitioned with the acquired batch-based boundary values.

During the training, for the first iteration, we directly use the boundary values derived from the first batch of data as the dynamic boundary values. Subsequently, since the second iteration, boundaries are updated adaptively in a momentum-based manner:

$$\boldsymbol{\nu}_t = \gamma \boldsymbol{\nu}_{t-1} + (1 - \gamma)\boldsymbol{\nu}_c \,, \tag{3}$$

where $\boldsymbol{\nu}_{t-1}$ stands for the bin partitioning boundaries used in the last iteration, and $\boldsymbol{\nu}_t$ is the updated dynamic boundaries used for the current iteration. $\gamma \in (0, 1)$ is the momentum update factor. With updated boundary values $\boldsymbol{\nu}_t$, points in each shape are divided into $n_b$ subsets of $\{\mathcal{B}_1, \mathcal{B}_2, \ldots, \mathcal{B}_{n_b}\}$ based on their sampling scores.

The principle idea presented here is the adaptive learning of boundary values, which are derived from the entirety of shapes within the training dataset. These values aim to evenly partition the distribution of point sampling scores across all shapes and points in the training data. Consequently, for each individual shape, the acquired boundary values can effectively partition its points into bins with a shape-specific strategy, capturing the unique characteristics of the shape while maintaining a degree of proximity to other shapes within the dataset.

**Tokens for Learning Bin Weights.** With points already being partitioned into bins for each shape, the next step is to learn a shape-specific sampling strategy, i.e., to learn shape-specific sampling weights for each bin. Inspired by ViTDosovitskiy et al. (2020), VilTKim et al. (2021), and Mask3DSchult et al. (2023) —which leverage additional tokens during the computation of attention maps to extract and convey information across the entire feature map or specific groups of points or pixels — we introduce additional tokens specifically for learning bin sampling weights. In our case, attention maps are computed shape-specific during the downsampling process, facilitating the learning of bin sampling weights also in a shape-specific manner.

Using the former proposed bin partitioning method, points in each shape are partitioned into $n_b$ subsets of $\{\mathcal{B}_1, \mathcal{B}_2, \ldots, \mathcal{B}_{n_b}\}$. The sampling weight $\omega_j$ for bin $\mathcal{B}_j (j = 1, 2, \ldots, n_b)$ is established based on the distinctive features of each shape. Fig. 5 gives the network structure of our proposed downsampling layer and illustrates the idea of using additional tokens. $n_b$ bin tokens are introduced during the attention computation, where each token corresponds to a specific bin. As shown in

Fig. 5, the bin tokens are initially concatenated with the input point-wise features for *Key* and *Value*. Subsequently, the combined features are subjected to a cross-attention mechanism with the original point-wise features as *Query*. The attention map is split into two parts of a point-to-point sub-attention map and a point-to-token sub-attention map. For the point-to-point attention map, the methods proposed in Sec. 3.1 and Sec. 3.2 are applied to it to obtain point-wise sampling scores. Note that in this case, the row-wise sum is not exactly equal to 1 but still very close to 1 since $n_b$ is of a very small quantity compared to $N$. With computed point scores, dynamic boundary values $\mathbf{v}_t$ are obtained for bin partitioning. Using the information regarding the allocation of points to respective bins, a mask operation is performed on the point-to-token sub-attention map as illustrated in Block B of Fig. 5. The sampling weights $\omega_j$ are then subsequently acquired with

$$\omega_j = \text{ReLU}(\frac{1}{\beta_j} \sum_{\mathbf{p}_i \in \mathcal{B}_j} m_{\mathbf{p}_i, \mathcal{B}_j}), \tag{4}$$

where $\beta_j$ stands for the number of points in bin $\mathcal{B}_j$, and $m_{\mathbf{p}_i, \mathcal{B}_j}$ represents the element in the energy matrix corresponding to point $\mathbf{p}_i$ in row and $\mathcal{B}_j$ in column.

**In-Bin Point Sampling.** For each shape, by considering the number of points contained within bins $\boldsymbol{\beta} = (\beta_1, \beta_2, \ldots, \beta_{n_b})$ alongside the determined bin sampling weights $\boldsymbol{\omega} = (\omega_1, \omega_2, \ldots, \omega_{n_b})$, the specific numbers of points to be selected from each bin $\boldsymbol{\kappa} = (\kappa_1, \kappa_2, \ldots, \kappa_{n_b})$ need to be determined. Direct multiplication of $\boldsymbol{\beta}$ and $\boldsymbol{\omega}$ does not yield a sum that aligns with the total number of down-sampled points $M$ required by the network structure. To address this discrepancy, a scaling method is applied to first scale bin sampling weights $\omega_j$. Furthermore, to prevent $\kappa_j$ from surpassing the available number $\beta_j$ in any bin, any excess points are proportionately redistributed to other bins that have not been fully sampled. The detailed pipeline is described in Algorithm 1.

Finally, within bin $\mathcal{B}_j$, $\kappa_j$ points are selected through random sampling with priors. The sampling probabilities $\rho_{\mathbf{p}_i}$ is determined by performing a softmax operation over the normalized point sampling score $a_{\mathbf{p}_i}$ with a temperature parameter $\tau$:

$$\rho_{\mathbf{p}_i} = \frac{e^{a_{\mathbf{p}_i}/\tau}}{\sum_{\mathbf{p}_i \in \mathcal{B}_j} e^{a_{\mathbf{p}_i}/\tau}}. \tag{5}$$

## 4 EXPERIMENTS

### 4.1 CLASSIFICATION

**Experiment Setting.** ModelNet40 classification benchmark Wu et al. (2015) contains 12,311 manufactured 3D CAD models in 40 common object categories. For a fair comparison, we use the official train-test split, in which 9,843 models are used for training and 2,468 models for testing. From each model mesh surface, points are uniformly sampled and normalized to the unit sphere. Only 3D coordinates are used as point cloud input. For data augmentation, we randomly scale, rotate, and shift each object point cloud in the 3D space. We use $n_b = 6$ bins for point partitioning. The momentum update factor $\gamma = 0.99$ for updating boundary values. The temperature parameter $\tau = 0.1$. More training details are provided in the Appendix.

**Qualitative and Quantitative Results.** Qualitative results of SAMBLE are presented in Fig. 6, including sampling score heatmaps, learned bin partitioning strategy with bin sampling ratios, and the final sampled results. From it, we can see that SAMBLE successfully samples enough edge points which construct the general structure of the shape. It also captures better global uniformity by not focusing heavily on edge points, especially for those thin/detailed parts (e.g. chair legs). From the logged shape bin histograms, we can see that shape-specific sampling strategies have been successfully learned. More visualization results are provided in the appendix, showcasing an intriguing pattern where shapes of the same category exhibit similar histogram distributions and sampling strategies. Overall, SAMBLE successfully achieves a better trade-off between sampling edge points and preserving global uniformity. Quantitative result is given in Tab. 2. Our method performs better than other methods and achieves state-of-the-art performance.

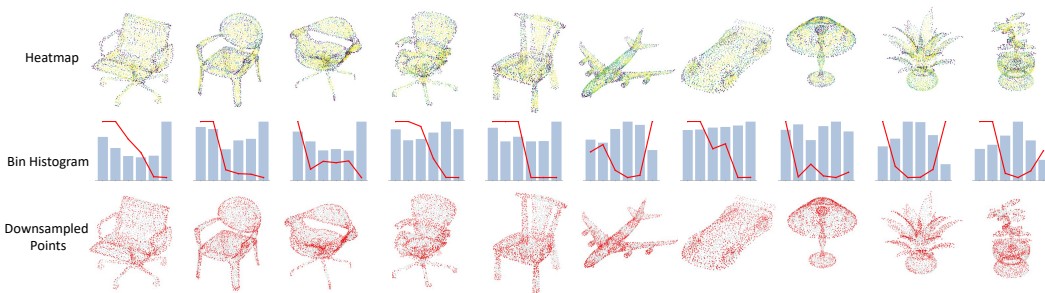

Figure 6: Qualitative results of our proposed SAMBLE. Apart from the sampled results, sampling score heatmaps and bin histograms along with bin sampling ratios are also given. All shapes are from the test set. Zoom in for optimal visual clarity.

Table 2: Numerical results on the ModelNet40 classification benchmark and the ShapeNet part segmentation benchmark.

| Method | Cls. OA (%) | Seg. Cat. mIoU (%) | Seg. Ins. mIoU (%) |
|---|---|---|---|
| PointNet++ | 91.9 | 81.9 | 85.1 |
| DGCNN | 92.9 | 82.3 | 85.2 |
| PointConv | 92.5 | 82.8 | 85.7 |
| PointTransformer | 93.7 | 83.7 | 86.6 |
| PointNeXt | 93.2 | 84.4 | **86.7** |
| PointMetaBase | - | 84.3 | **86.7** |
| APES (local) | 93.5 | 83.1 | 85.6 |
| APES (global) | 93.8 | 83.7 | 85.8 |
| SAMBLE | **94.2** | **84.5** | 86.7 |

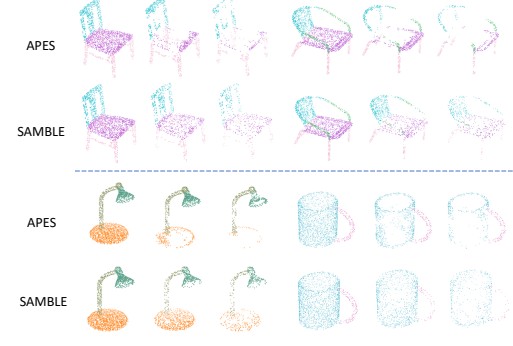

Figure 7: Segmentation results of our proposed SAMBLE. All shapes are from the test set.

## 4.2 SEGMENTATION

**Experiment setting.** The ShapeNetPart dataset Yi et al. (2016) is used for 3D object part segmentation. It consists of 16,880 models from 16 shape categories, with 14,006 3D models for training and 2,874 for testing. The number of parts for each category is between 2 and 6, with 50 different parts in total. We use the sampled point sets produced in Qi et al. (2017a) for a fair comparison with prior work. For evaluation metrics, we report category mIoU and instance mIoU. We use $n_b = 4$ bins for point partitioning. The momentum update factor $\gamma = 0.99$ for updating boundary values. The temperature parameter $\tau = 0.1$. More training details are provided in the Appendix.

**Qualitative and Quantitative Results.** Qualitative results are presented in Fig. 7. From it, we can observe that compared to APES which focuses heavily on edge points, our SAMBLE strikes a better balance between sampling edge points and shape global uniformity. For example, SAMBLE exhibits a more balanced utilization of non-edge points, as exemplified by the chair seat. It demonstrates a thoughtful sampling strategy that takes into account different point categories, resulting in a more comprehensive representation of the shape. Quantitative results are provided in Tab. 2, which shows that our SAMBLE achieves state-of-the-art performance.

For the part segmentation benchmark, we further report the performance on the intermediate downsampled sub-point clouds in Tab. 3. Additionally, results from PointNeXt Qian et al. (2022) are also presented, which is a prominent point cloud learning method that employs FPS for downsampling. It is evident that FPS-based methods exhibit poorer performance when applied to intermediate downsampled sub-point clouds. In contrast, our SAMBLE approach demonstrates improved performance with intermediate downsampled sub-point clouds, showing the superiority of our proposed sampling methods.

Table 3: Segmentation performances on intermediate downsampled point clouds.

| Method | PointNeXt | | | SAMBLE | | |
|---|---|---|---|---|---|---|
| Point Number | 2048 | 1024 | 512 | 2048 | 1024 | 512 |
| Cat. mIoU (%) | 84.40 | 83.79 | 82.77 | 84.51 | 84.84 | **85.04** |
| Ins. mIoU (%) | 86.70 | 86.18 | 85.18 | 86.67 | 86.93 | **87.12** |

Table 4: Comparison with other sampling methods. Evaluated on the ModelNet40 classification benchmark with multiple sampling sizes.

| $M$ | Voxel | RS | FPS | S-NET | SampleNet | MOPS-Net | LighTN | APES (w/ pre-pro.) | APES (w/o pre-pro.) | SAMBLE |
|---|---|---|---|---|---|---|---|---|---|---|
| 512 | 73.82 | 87.52 | 88.34 | 87.80 | 88.16 | 86.67 | 89.91 | **90.81** | 89.81 | 90.58 |
| 256 | 73.50 | 77.09 | 83.64 | 82.38 | 84.27 | 86.63 | 88.21 | **90.40** | 86.78 | 90.18 |
| 128 | 68.15 | 56.44 | 70.34 | 77.53 | 80.75 | 86.06 | 86.26 | 89.77 | 84.87 | **90.02** |
| 64 | 58.31 | 31.69 | 46.42 | 70.45 | 79.86 | 85.25 | 86.51 | 89.57 | 79.23 | **89.81** |
| 32 | 20.02 | 16.35 | 26.58 | 60.70 | 77.31 | 84.28 | 86.18 | 88.56 | 75.63 | **89.45** |

## 4.3 FEW-POINT SAMPLING

**Experiment setting.** We additionally compare our sampling method to previous work including RS, FPS, and the more recent learning-based S-Net, SampleNet, LighTN, APES, etc. The same evaluation framework from Dovrat et al. (2019); Wang et al. (2023); Wu et al. (2023a) is used. The point cloud is first sampled into a limited number of points, and subsequently the downsampled result is fed into a task network for evaluation. The task here is the ModelNet40 Classification, and the task network is PointNet. All sampling methods are evaluated with multiple sampling sizes.

**Qualitative and Quantitative Results.**
Quantitative results are presented in Tab. 4. Note that APES Wu et al. (2023a) uses FPS to pre-process the input into $2M$ points while we did not. For a fair comparison, additional results of APES without the pre-processing step are also tested and reported. Nonetheless, even without pre-processing, SAMBLE achieves state-of-the-art results in the few-point sampling task as the number of sampled points decreases to smaller ones.

Qualitative results are presented in Fig. 8. For few-point sampling, APES relies on FPS to pre-sample the input into $2M$ points due to its limitations . In contrast, our method preserves better global uniformity, allowing direct few-point sampling

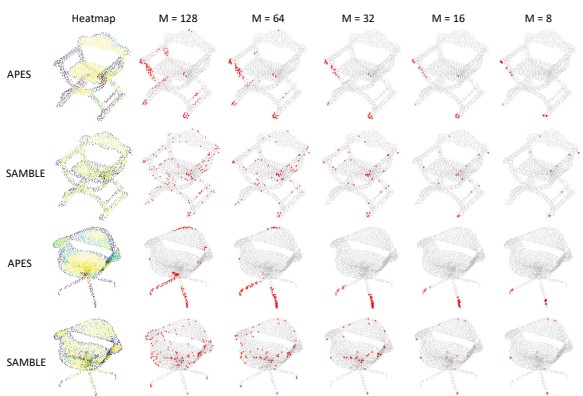

Figure 8: Sampled results of few-point sampling in comparison with APES. Zoom in for optimal clarity.

from the input while still achieving satisfactory sampled results, as demonstrated in Fig. 8. When sampling very few points, APES tends to concentrate on the sharpest regions, whereas our SAMBLE method preserves better global uniformity throughout the point cloud shape.

## 4.4 ABLATION STUDY

In this subsection, our emphasis is directed toward the novel designs introduced within this paper, excluding common topics such as network width. More ablation study and further design justifications are provided in the appendix to enhance the interpretability of our proposed method.

**Different Indexing Modes.** Apart from the visualized heatmaps given in Fig. 4, we also report their respective experimental results in Tab. 5. The tests are performed using top-M as the sampling strategy. From it, we can observe that indexing modes vi and vii achieve relatively best performances.

Table 5: Classification and segmentation performance with different indexing modes.

| | Indexing Mode | i | ii | iii | iv | v | vi | vii |
|---|---|---|---|---|---|---|---|---|
| Cls. | OA (%) | 93.92 | 93.78 | 93.63 | 93.66 | 93.40 | **94.11** | 94.08 |
| Seg. | Cat. mIoU (%) | 83.98 | 83.85 | 83.62 | 83.51 | 83.47 | 84.12 | **84.22** |
| | Ins. mIoU (%) | 86.16 | 85.99 | 85.74 | 85.60 | 85.49 | 86.38 | **86.46** |

**Number of Bins.** As a key parameter in SAMBLE, an ablation study is performed over the number of bins $n_b$. The results are presented in Tab. 6. Remarkably, increasing the number of bins does not yield improved performance. This phenomenon is likely attributable to the subdivision of shapes into an excessive number of point categories, leading to the gradual diminishment of score disparities across the bins. In our case, $n_b = 6/4$ yields the best performance for the classification and segmentation tasks respectively, and we use it for the corresponding experiments.

Table 6: Classification and segmentation performance with different number of bins.

| Number of Bins | | 1 | 2 | 4 | 6 | 8 | 10 | 12 |
|---|---|---|---|---|---|---|---|---|
| Cls. | OA (%) | 94.05 | 93.91 | 93.98 | **94.18** | 94.02 | 93.80 | 93.84 |
| Seg. | Cat. mIoU (%) | 84.22 | 84.14 | **84.51** | 84.40 | 84.19 | 83.98 | 84.36 |
| | Ins. mIoU (%) | 86.46 | 86.28 | **86.67** | 86.61 | 86.48 | 86.23 | 86.43 |

**Upsampling layer.** An important aspect to highlight is the upsampling layer. Most point cloud network models employ neighbor-based interpolation Qi et al. (2017b); Zhao et al. (2021); Qian et al. (2022) for upsampling, as FPS is typically used during the downsampling process. However, APES introduces a cross-attention layer for upsampling to address the limitations of overemphasizing edge points, which renders traditional neighbor-based interpolation impractical. In contrast, our method strikes a better balance between sampling edge points and maintaining global uniformity, allowing the use of interpolation operations during upsampling. An ablation study for evaluating various upsampling layers and interpolation with different $K_{up}$ values is conducted, and the results are presented in Table 7. The results show a performance drop for APES when interpolation is used in place of cross-attention, while SAMBLE demonstrates superior performance with interpolation.

Table 7: Segmentation results with different upsampling layers on ShapeNet Part. The number before "/" is the category mIoU, and the number after is the instance mIoU.

| Upsample | Interpolation | | | Cross-Attention |
|---|---|---|---|---|
| | $K_{up} = 3$ | $K_{up} = 8$ | $K_{up} = 16$ | |
| APES (local) | 82.89 / 85.40 | 82.95 / 85.44 | 82.96 / 85.42 | 83.11 / 85.58 |
| APES (global) | 83.16 / 85.53 | 83.19 / 85.59 | 83.17 / 85.55 | 83.67 / 85.81 |
| SAMBLE | **84.51 / 86.67** | 84.35 / 86.48 | 84.31 / 86.43 | 84.36 / 86.44 |

## 5 CONCLUSION

In this paper, a new point cloud sampling method has been proposed to learn shape-specific sampling strategies for achieving better trade-off between sampling local details and preserving global uniformity. Based on a sparse attention map that combines the knowledge from both local and global information, multiple indexing modes have been designed and explored. By partitioning the points in each shape into bins, and learning respective sampling ratios for each bin with additional tokens, shape-specific sampling strategies are acquired for individual point cloud shapes. With the proposed methods, we achieve a more effective balance between capturing local details and preserving global uniformity of the input shape, resulting in improved performance on downstream tasks.

Looking forward, the trade-off between sampling local details and preserving global uniformity in point clouds remains an open challenge. Future advancements in upsampling layers could further benefit from leveraging previously discarded information to refine this balance. The complex interaction between downsampling and upsampling layers presents a promising area for further research. Another exciting direction is adapting the proposed method to point cloud scenes rather than isolated shapes. This shift introduces the challenge of scene boundary points being mistakenly prioritized as significant, which calls for more sophisticated sampling algorithms. Additionally, the proposed approach could be extended to other 3D data representations, such as 3D Gaussian Splatting, where each point is represented as a 3D Gaussian. Given the typically large size of such 3D data, introducing effective sampling techniques could significantly enhance its processing efficiency.

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

# Appendix

## A  NETWORK ARCHITECTURE

For a fair comparison, the same basic network architectures from APES are used in our experiments, as illustrated in Fig. 9. The downsampling layers are replaced with our proposed ones, and the upsampling layers are replaced with the classical interpolation-based ones.

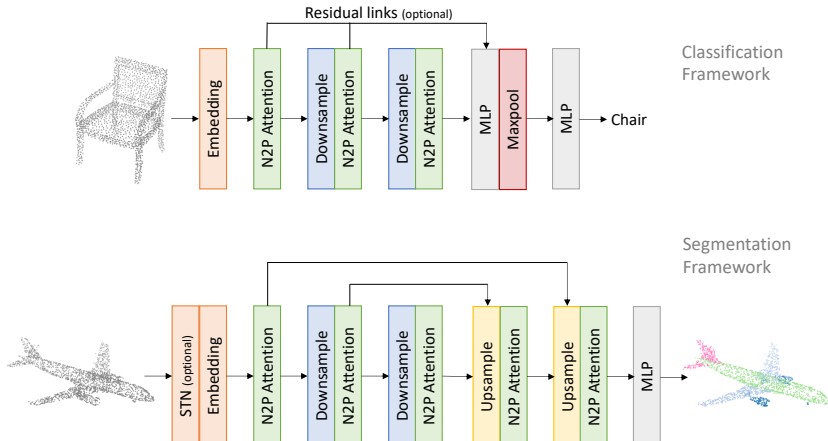

Figure 9: Network architectures for the classification task and the segmentation task.

## B  MORE TRAINING DETAILS

**Classification Tasks.** AdamW is used as the optimizer. The learning rate starts from $1 \times 10^{-4}$ and decays to $1 \times 10^{-8}$ with a cosine annealing schedule. The weight decay hyperparameter for network weights is set as 1. Dropout with a probability of 0.5 is used in the last two fully connected layers. We use $n_b = 6$ bins for point partitioning. The momentum update factor $\gamma = 0.99$ for updating boundary values. The temperature parameter $\tau = 0.05$. The network is trained with a batch size of 8 for 200 epochs.

**Segmentation Tasks.** AdamW is used as the optimizer. The learning rate starts from $1 \times 10^{-4}$ and decays to $1 \times 10^{-8}$ with a cosine annealing schedule. The weight decay hyperparameter for network weights is $1 \times 10^{-4}$. We use $n_b = 4$ bins for point partitioning. The momentum update factor $\gamma = 0.99$ for updating boundary values. The temperature parameter $\tau = 0.05$. The network is trained with a batch size of 16 for 200 epochs.

# C   SAMPLING RESULTS IN COMPARISON WITH APES

Additional qualitative results in comparison with APES are provided in Fig. 10 and Fig. 11. Both figures indicate that APES focuses too heavily on edge points, while SAMBLE successfully achieves a better balance between sampling edge points and preserving global uniformity, leading to better performance on downstream tasks.

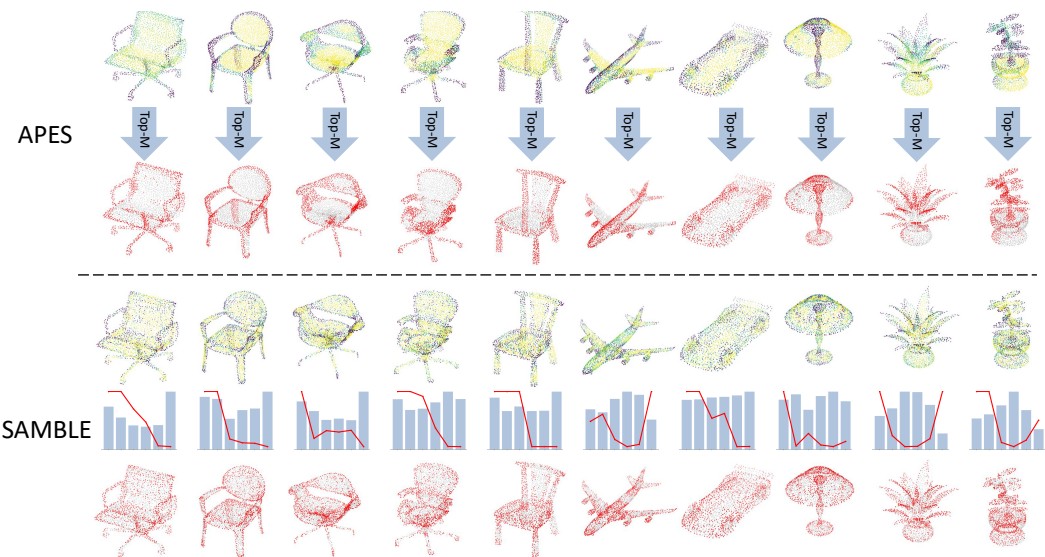

Figure 10: Qualitative results of our proposed SAMBLE, in comparison with APES. In addition to the sampled results, sampling score heatmaps and sampling strategies are also provided.

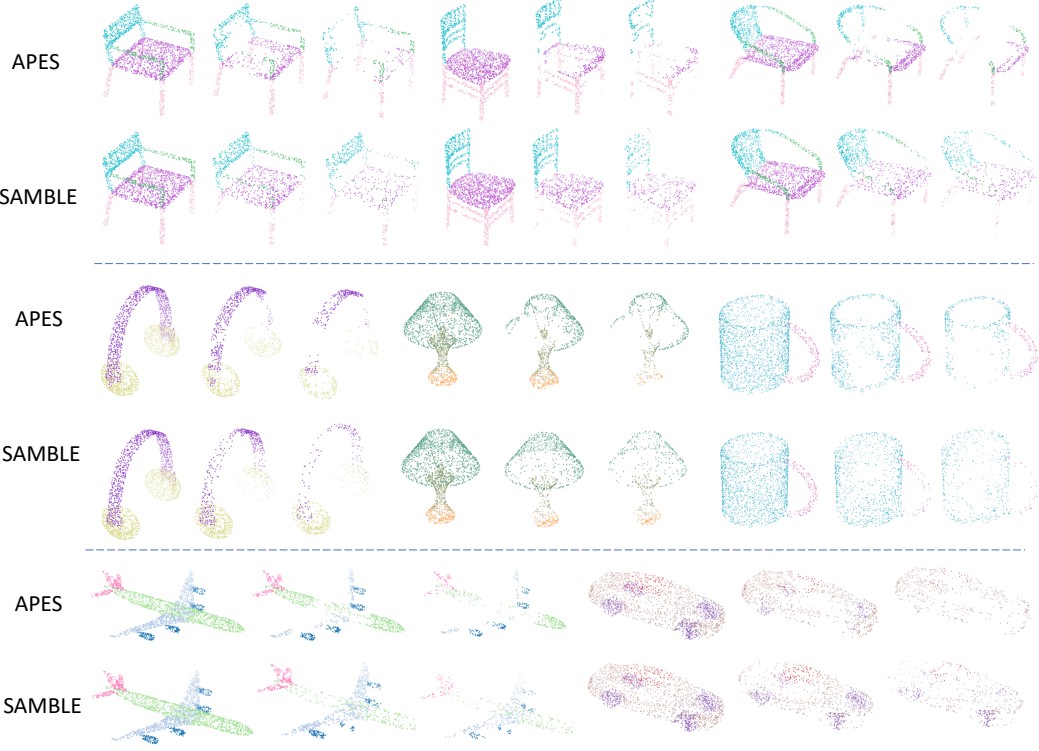

Figure 11: Segmentation results of our proposed SAMBLE, in comparison with APES.

# D    SAMPLING POLICIES.

An illustration of different sampling policies is provided in Fig. 12, including Top-M sampling, prior-based sampling, and bin-based sampling. The Top-M sampling policy samples the points with larger sampling scores directly. The prior-based sampling policy samples points randomly according to their converted sampling probabilities. The bin-based sampling policy further builds upon that. It first partitions the point set into several bins, and then samples points within each bins. In each bin, either top-M sampling or prior-based sampling can employed. In our case, we use the prior-based sampling. The bin-based sampling policy allows for more fine-grained control over the sampling process, tailoring it to the specific characteristics of each shape.

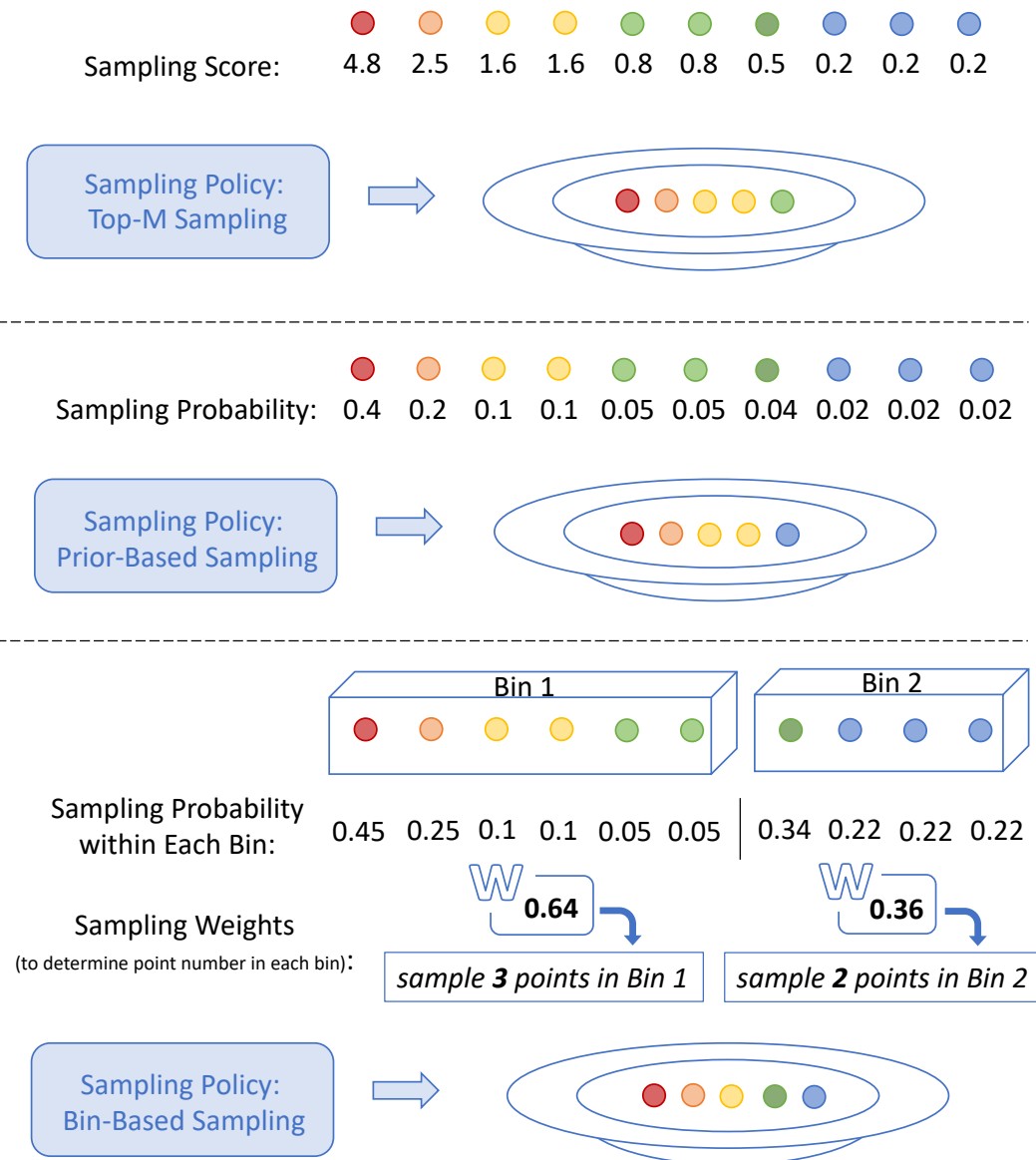

Figure 12: An illustration of different sampling policies. Note for bin-based sampling, either top-M sampling or prior-based sampling may be used within each bin.

# E    RELATIONSHIP BETWEEN BIN SAMPLING WEIGHTS AND RATIOS

For the sake of brevity and improved visual clarity, in the paper, the axis labels of the histograms have been omitted. We further provide the full version of the histogram, in which the number of points and the sampling ratio in each bin are given. A demo is provided in Fig. 13. More detailed histogram results are provided in Sec. I.

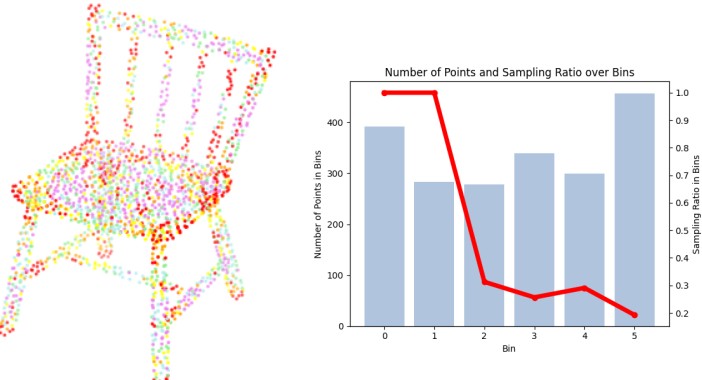

Figure 13: Left: bin partitioning, each color represents the points belonging to this bin. Right: the learned sampling strategy.

One thing worth noting is that the indicated sampling ratios $\mathbf{r}$ in the histogram are not simply re-scaled sampling weights $\boldsymbol{\omega}$. As in the algorithm we presented in the paper, apart from the re-scaling operation, a redistribution operation is also applied to prevent $\kappa_j$ surpassing the available point number $\beta_j$ in one bin. Given the point number in each bin $\boldsymbol{\beta} = (\beta_1, \beta_2, \ldots, \beta_{n_b})$ and the number of points to be sampled from each bin $\boldsymbol{\kappa} = (\kappa_1, \kappa_2, \ldots, \kappa_{n_b})$, the sampling ratios presented in the histogram is $\mathbf{r} = \boldsymbol{\kappa}/\boldsymbol{\beta}$ and $\mathbf{r} \in [0, 1]$.

The redistribution operation only happens when $\kappa_j$ is about to surpass $\beta_j$, this means all points in $j$th bin have been selected and $r_j = 1$. We additionally count and document the likelihood of this occurrence for all bins across all test shapes. The numbers are reported in Tab. 8, for which we can see that for around 54% of the shapes, all points in the first bin are selected and sampled. Note that the first bin corresponds to the points of higher sampling scores which are mostly edge points with indexing mode vii. This observation underscores the significance of edge points. On the other hand, there are still around 46% shapes that do not sample all edge points. It suggests that an excessive emphasis on edge points might have adverse effects on subsequent downstream tasks, which also aligns with the conclusion drawn by APES.

Table 8: Possibilities of all points being sampled in bins, across all test shapes.

| Bin Index | 0 | 1 | 2 | 3 | 4 | 5 |
|---|---|---|---|---|---|---|
| Possibilities of All Points Being Sampled | 53.69% | 27.11% | 8.02% | 2.11% | 0.85% | 4.98% |

# F    DESIGN JUSTIFICATIONS OF THE BIN TOKEN IDEA - DEVIL IS IN THE DETAILS.

**Adding Bin Tokens to Q or K/V?** A critical point in the idea of bin tokens lies in determining the specific branches to which the tokens should be concatenated. In order to match the tensor dimension for later computation in the attention mechanism, the tensor size of Key and Value should be the same. Hence if tokens are being added to the Key branch, they also have to be added to the Value branch. Overall, there are two possibilities of adding bin tokens to (i) the Query branch, or (ii) the Key and the Value branches.

It is crucial to emphasize that, due to the nature of the sampling operation where indexes are selected, gradients cannot be propagated back through the sampling operation during the backward propagation process. As a result, regardless of the selected structure, it is essential to establish an alternative pathway to convey the information contained within the bin tokens, which have a size of $n_b \times N$, to the downsampled features, which have a size of $M \times d$. This pathway should ensure the flow of relevant information despite the inability to directly backpropagate gradients through the sampling operation.

As illustrated in the left of Fig. 14, in the former case, an attention map of tensor size $(N + n_b) \times N$ is obtained. After $M$ indexes of the points to be sampled are learned with SAMBLE, $M$ rows in the attention map are extracted to form a new tensor for the next steps. However, note that the sub-tensor of $n_b \times N$ will never be delivered to the next steps since they do not correspond to points, hence no gradient will be backpropagated to the tokens during the training.

### Adding Bin Tokens to *Query*   Adding Bin Tokens to *Key* and *Value*

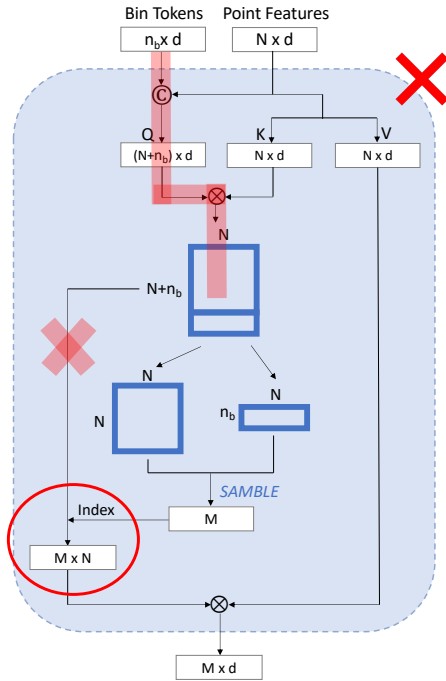 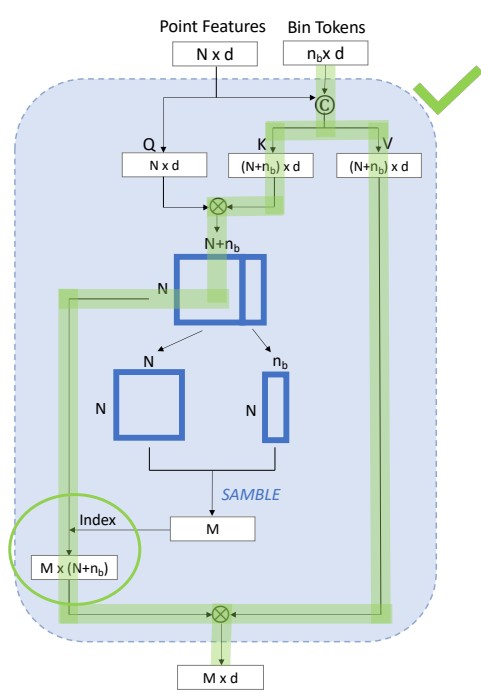

Figure 14: Adding bin tokens to Query leads to no gradient being backpropagated to the tokens, while adding bin tokens to Key and Value enables the gradient backpropagation.

On the other hand, as illustrated in the right of Fig. 14, adding bin tokens to the Key and Value branches does not have this problem and successfully enables gradient backpropagation. One thing worth mentioning is that in this scenario, the row-wise sum is not exactly equal to 1 but still very close to 1 due to the significantly smaller magnitude of $n_b$ relative to $N$. Therefore, this is unlikely to significantly impact the calculation of point-wise sampling scores. Concerning the design of adding bin tokens to all branches of Query, Key, and Value, it is equivalent to case ii since the sub-tensor of $n_b$ rows in the attention map will never be sampled and propagated.

**Order of Mean-pooling and ReLU Operations.** Within our design, the ReLU operation is used to prevent the learned sampling weight from being negative. It can be performed after Mean-pooling, as shown in Eq. 4, or performed before Mean-pooling:

$$\omega_j = \frac{1}{\beta_j} \sum_{\mathbf{p}_i \in \mathcal{B}_j} \text{ReLU}(m_{\mathbf{p}_i, \mathcal{B}_j}). \tag{6}$$

However, the inherent distribution of values within tensors often results in a non-negligible proportion being negative, especially those corresponding to points of lower importance. Directly setting

too many values to zero would result in a significant loss of features, which is regrettable considering the potential information discarded. Therefore, instead of performing the ReLU operation before the mean-pooling operation, we do it the other way around, i.e., first mean-pooling, then, after this information fusion, ReLU is performed over the pooled results.

Fig. 15 gives the learned sampling strategies with the mean-pooling and ReLU operations applied in different orders. Although both orders yield shape-specific sampling strategies, the sampling ratios over bins learned with the order of ReLU first are mostly around 40% - 60%, leading to a worse sampling performance. On the other hand, the order of mean-pool first yields better sampling strategies as less potential information is discarded.

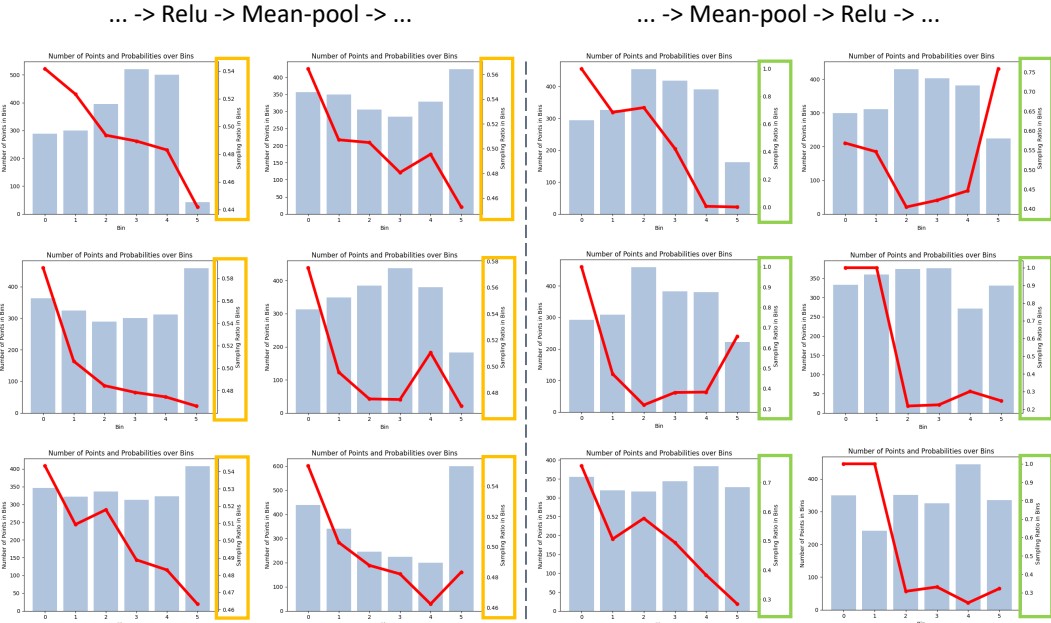

Figure 15: Learned sampling strategies with the mean-pooling and ReLU operations applied in different orders.

We additionally count and document the likelihood of ReLU being effective, which indicates the former pooled result is negative, for all bins across all test shapes. From the numbers reported in Tab. 9, we can see that the likelihood of the pooled results being negative is extremely small (less than 1%) for the first half of bins, while it goes higher for the latter bins yet the number is still relatively acceptable.

Table 9: Possibilities of ReLU being effective in bins, across all test shapes.

| Bin Index | 0 | 1 | 2 | 3 | 4 | 5 |
|---|---|---|---|---|---|---|
| Possibilities of ReLU Being effective | 0.45% | 0.28% | 0.57% | 4.25% | 11.63% | 13.53% |

**Pre-softmax or Post-softmax Attention Map for Splitting The Point-to-Token Sub-Attention Map.** When addressing the bin tokens, our initial approach involved splitting the point-to-token sub-attention map from the post-softmax attention map $M_{post}$, which seemed intuitively appropriate. Furthermore, all elements within $M_{post}$ are inherently positive, eliminating any concern for negative sampling weights and obviating the need for an additional ReLU operation. However, experimental findings revealed that this method proved ineffective, as it resulted in overly uniform sampling weights across different bins.

The underlying cause of this issue was identified after we explored the underlying mathematical principles and examined the values in the tensors during runtime. Tensors in a well-trained network

tend to exhibit diminutive feature values as they propagate through layers. Denote $m_{ij}$ as one element in the pre-softmax attention map $\mathbf{M}_{\text{pre}}$, given its minute magnitude, we apply the Taylor expansion formula to yield:

$$e^{m_{ij}} = 1 + m_{ij} + \frac{m_{ij}^2}{2} + \cdots \approx 1 + m_{ij} \,. \tag{7}$$

Therefore, the corresponding element $m'_{ij}$ in the post-softmax attention map is

$$m'_{ij} = \frac{e^{m_{ij}}}{\sum_{j=1}^{N+n_b} e^{m_{ij}}} \approx \frac{1 + m_{ij}}{N + n_b + \sum_{j=1}^{N+n_b} m_{ij}} \,. \tag{8}$$

In our case, the values of the elements $m_{ij}$ in $\mathbf{M}_{\text{pre}}$ are approximately within the magnitude of $10^{-3}$ to $10^{-5}$. After a softmax operation, the resultant values $m'_{ij}$ in $\mathbf{M}_{\text{post}}$ exhibit minimal variation, leading to closely similar sampling weights across bins in a later step.

Efforts were undertaken to address this issue before we turned to using $\mathbf{M}_{\text{pre}}$ for sampling weights acquisition. We attempted to use the logarithmic operation to restore the lost information:

$$\ln(m'_{ij}) = \ln\left(\frac{e^{m_{ij}}}{\sum_{j=1}^{N+n_b} e^{m_{ij}}}\right) = m_{ij} - \ln\left(\sum_{j=1}^{N+n_b} e^{m_{ij}}\right) \tag{9}$$

After the logarithmic operation, every value in the sub-attention map is negative. Therefore, a normalization operation is necessary. However, as shown in Fig. 16, the common normalization methods, such as z-score and centering, will result in too many negative elements (more than half), leading to too much information loss when passing through subsequent ReLU modules. Even if we successfully identify or meticulously design a superior normalization method that enables manual control over the proportion of negative elements to an applicable value, such manual intervention strays from the original intention of this thesis, which is to discover a learning-based mapping from sampling score to sampling probability.

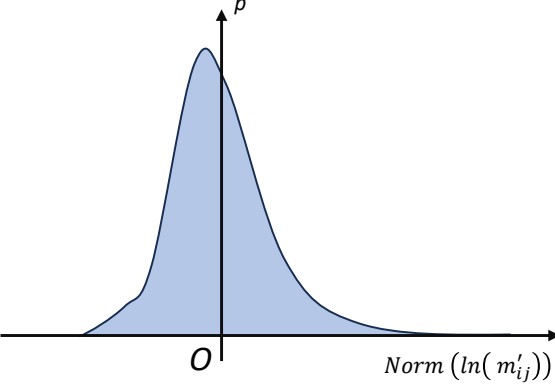

Figure 16: Illustrative figure of the distribution of the element values in the post-softmax attention map, after normalization.

Through the analysis, we observed that the term $m_{ij}$ in Eq. 9 is exactly the elements in the pre-softmax attention map and is what we are interested in. Therefore, to avoid the potential loss of information that could arise from the softmax operation, we opted to directly use the results from $\mathbf{M}_{\text{pre}}$ for bin sampling weights acquisition.

## G   ADDITIONAL ABLATION STUDIES

**Momentum Update Factor.** The momentum update strategy is widely used within contrastive learning frameworks in self-supervised learning. In our case, we aim to derive the bin boundary values $\boldsymbol{\nu}$ from the entirety of shapes within the training dataset. These values aim to evenly partition

the distribution of point sampling scores across all shapes and points in the training data. Hence such an adaptive learning method is used.

An ablation study over the momentum update parameter $\gamma$ is performed and the numerical results are reported in Tab. 10. From it, we can see that $\gamma = 0.99$ yields the best performance. This actually aligns with most current contrastive learning frameworks, where a majority use a value of $\gamma = 0.99$.

Table 10: Classification performance with different values of the momentum update factor $\gamma$.

|  | $\gamma$ | 0.9 | 0.99 | 0.999 | 0.9999 |
|---|---|---|---|---|---|
| Cls. | OA (%) | 93.80 | **94.18** | 94.02 | 93.95 |

We additionally provide the bin partitioning results over the test dataset with the learned boundary values $\boldsymbol{\nu}$ in Fig. 17. It demonstrates that the boundary values adaptively learned from the training dataset can also effectively partition the distribution of point sampling scores evenly across all shapes and points in the test dataset.

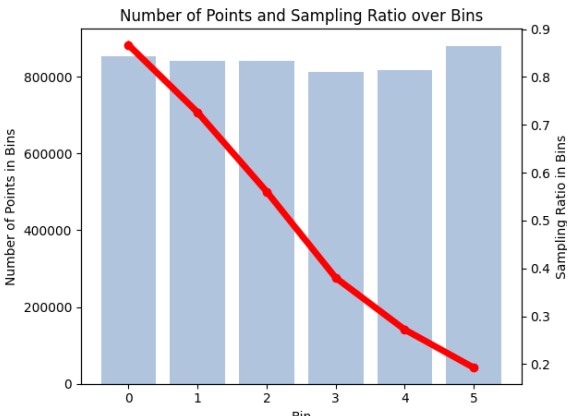

Figure 17: Partitioning the distribution of point sampling scores of all shapes and points in the test dataset into bins with the learned boundary values.

**Temperature Parameter.** The sampling strategy is determined with the point number in each bin $\boldsymbol{\beta} = (\beta_1, \beta_2, \dots, \beta_{n_b})$ and the number of points to be sampled from each bin $\boldsymbol{\kappa} = (\kappa_1, \kappa_2, \dots, \kappa_{n_b})$. Within each bin, instead of applying the top-M sampling method simply, we suggest employing random sampling with priors. The idea is quite straightforward: process the point-wise sampling scores into point-wise sampling probabilities, and $M$ non-repeated points are sampled randomly based on their sampling probabilities:

$$\rho_{\mathbf{P}_i} = \frac{e^{a_{\mathbf{P}_i}/\tau}}{\sum_{i=1}^{N} e^{a_{\mathbf{P}_i}/\tau}}, \tag{10}$$

where the temperature parameter $\tau$ controls the distribution of the sampling probabilities.

Within each bin, when $\tau$ is set close to 0, the sampling result would be close to top-M; When $\tau$ is set close to $+\infty$, the sampling result would be close to uniform sampling; when $\tau = 1$, the sampling result would be identical to the Softmax-based sampling. Hence, by manipulating this parameter, we can tune the sampling process from uniform sampling, to the conventional Softmax-based sampling, and further to the top-M sampling.

An ablation study over the value of $\tau$ has been conducted. To better illustrate this idea, the pre-softmax point sampling score heatmap and the post-softmax point sampling probability heatmap are visualized in Fig. 18. However, please note that since the softmax operation is performed within each bin, it would be impossible to visualize the post-softmax sampling probabilities of different bins in a same figure if multi-bins are used. Hence in Fig. 18 only a single bin is used, i.e. $n_b = 1$. From it, we can observe that the sampling probability of points goes from having a large deviation

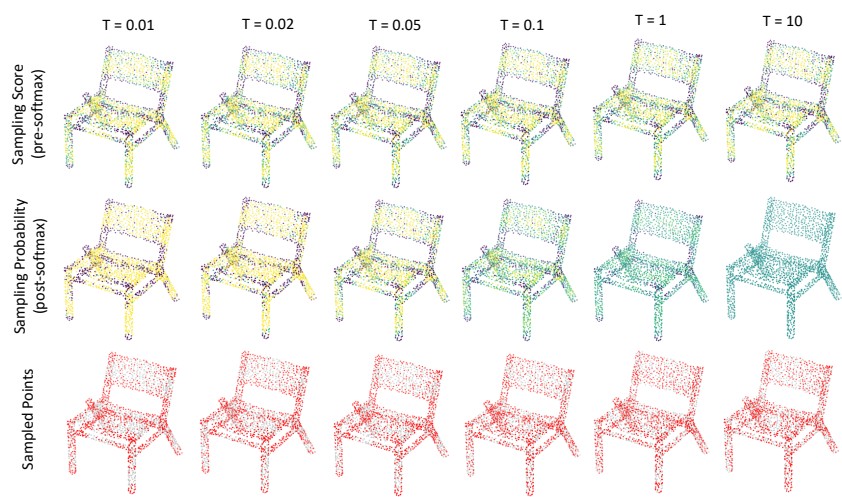

Figure 18: Different sampling results using different $\tau$ in the softmax with temperature during the sampling process. The indexing mode is the sparse column square-divided.

Table 11: Classification and segmentation performance of the model with different $\tau$ values.

| | $\tau$ | 0.01 | 0.02 | 0.05 | 0.1 | 0.2 | 0.5 | 1 | 10 |
|---|---|---|---|---|---|---|---|---|---|
| Cls. | OA (%) | 93.84 | 93.96 | 94.06 | **94.18** | 93.89 | 93.84 | 93.74 | 93.70 |
| Seg. | Cat. mIoU (%) | 84.10 | 84.23 | 84.38 | **84.51** | 84.26 | 84.13 | 84.02 | 83.88 |
| | Ins. mIoU (%) | 86.44 | 86.48 | 86.60 | **86.67** | 86.51 | 86.42 | 86.29 | 86.23 |

to being uniformly distributed, just as we designed. Numerical results are reported in Tab. 11, where $\tau = 0.1$ achieves the best performance. Moreover, a smaller $\tau$, which leads to a sampling strategy close to Top-M, does not always guarantee better performance. This is consistent with the conclusion that sampling only edge points can be detrimental.

## H  MODEL COMPLEXITY

To evaluate SAMBLE's practicality, we assess its complexity in comparison with APES and report the results in Tab. 12. This includes details on model parameters and FLOPs for both the entire model and a single downsampling layer. In order to assess inference efficiency, experiments were carried out using a trained ModelNet40 classification model on a single NVIDIA GeForce RTX 3090. The tests were conducted with a batch size of 8, evaluating a total number of 2468 shapes from the test set.

Table 12: For model complexity, we report the number of parameters and FLOPs for both full model and one downsampling layer. The inference throughput (instances per second) is also reported.

| Method | Params. | | FLOPs | | Throughput (ins./sec.) |
|---|---|---|---|---|---|
| | Full Model | One DS Layer | Full Model | One DS layer | |
| APES (local) | 4.47M | 49.15k | 4.59G | 1.09G | 488 |
| APES (global) | 4.47M | 49.15k | 3.03G | 0.05G | 520 |
| SAMBLE ($n_b = 1$) | 4.47M | 49.15k | 3.03G | 0.05G | 473 |
| SAMBLE ($n_b = 6$) | 4.48M | 66.56k | 3.56G | 0.38G | 125 |

As shown in Tab. 12, SAMBLE has a slightly larger number of model parameters compared to APES, primarily due to the incorporation of additional bin tokens. Notably, when $n_b = 1$, the number of parameters and FLOPs of SAMBLE are identical to that of APES. This is quite reasonable as in this case, using additional bin tokens is unnecessary and the multi-bin-based sampling policy

degrades into the simple prior-based sampling policy. On the other hand, SAMBLE's inference throughput is reduced due to the introduction of bin partitioning operations. Notably, the process of determining the number of points to be sampled within each bin involves a CPU-intensive loop computation, which can lead to increased inference time.

# I  MORE VISUALIZATION RESULTS OF LEARNED SHAPE-SPECIFIC SAMPLING STRATEGIES

We present additional extensive results in Fig. 19, Fig. 20, Fig. 21, and Fig. 22 with various categories. From them, we can observe that shape edge points are mostly partitioned into the first two bins. Furthermore, in addition to learning shape-wise sampling strategies for individual shapes, it is observed that analogous shapes within the same category exhibit similar histogram distributions and sampling strategies. Conversely, point clouds from different shape categories are sampled by distinct sampling strategies.

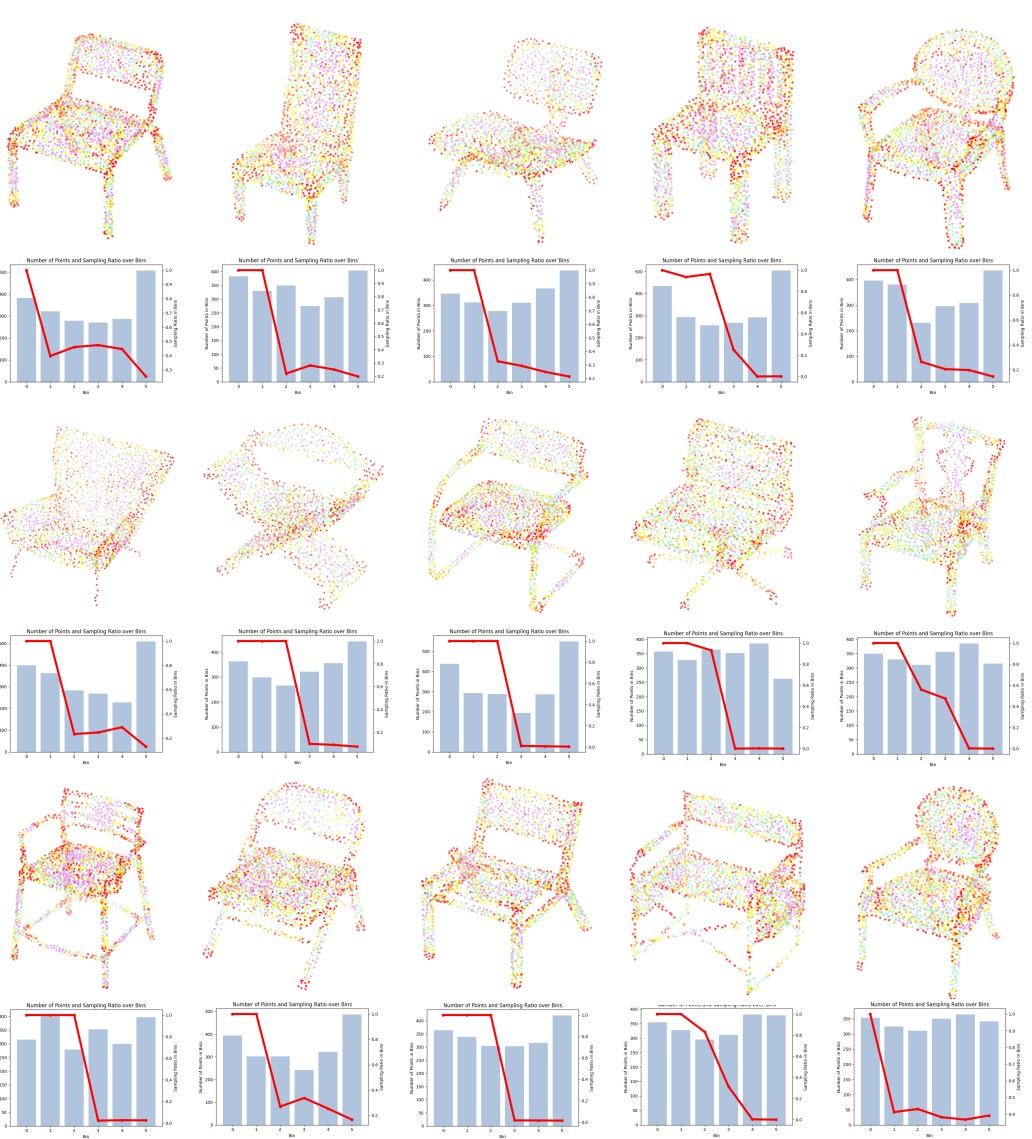

Figure 19: More visualization results of bin partitioning and learned shape-specific sampling strategies. The chair category. Zoom in for optimal visual clarity.

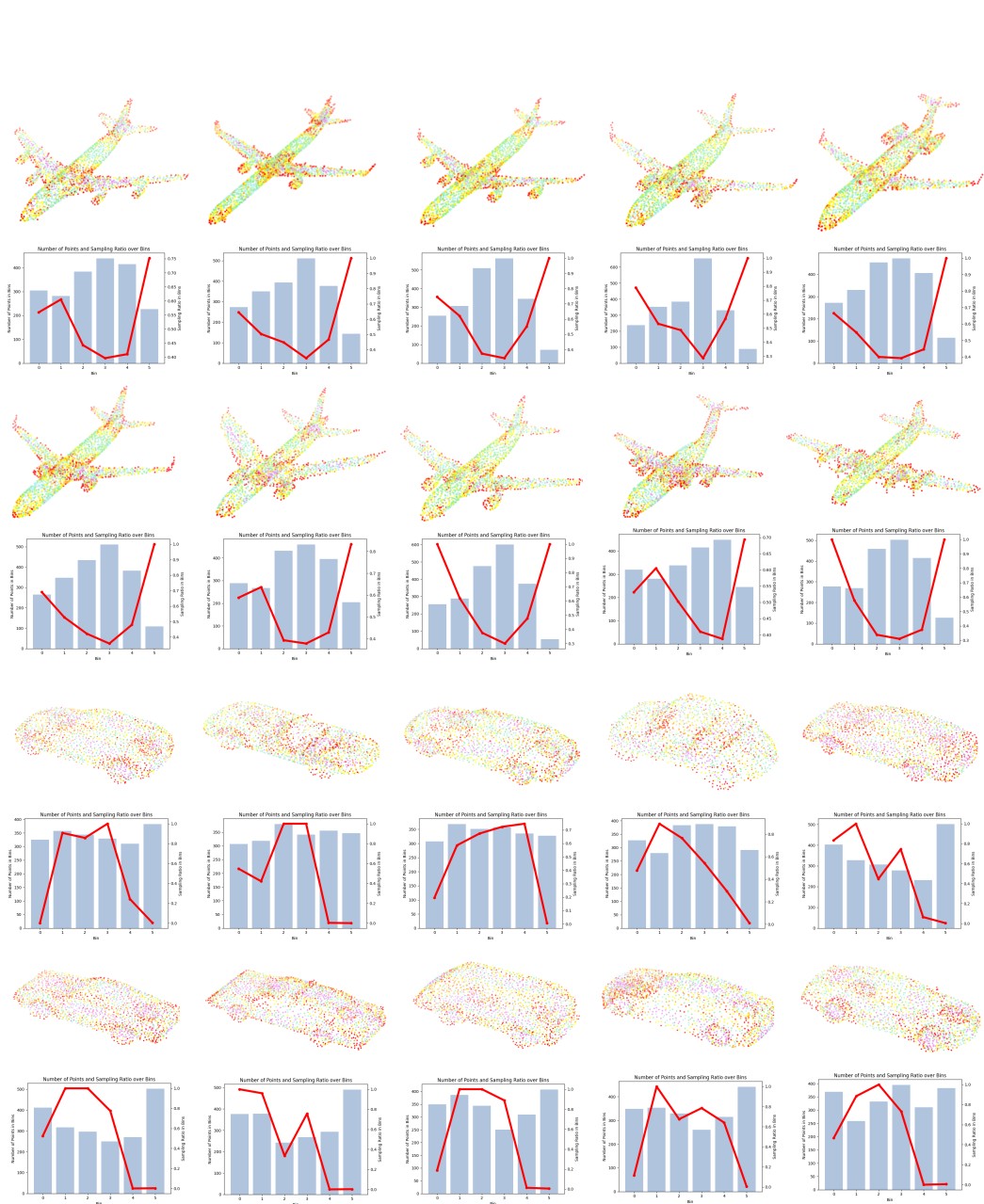

Figure 20: More visualization results of bin partitioning and learned shape-specific sampling strategies. The airplane and car categories. Zoom in for optimal visual clarity.

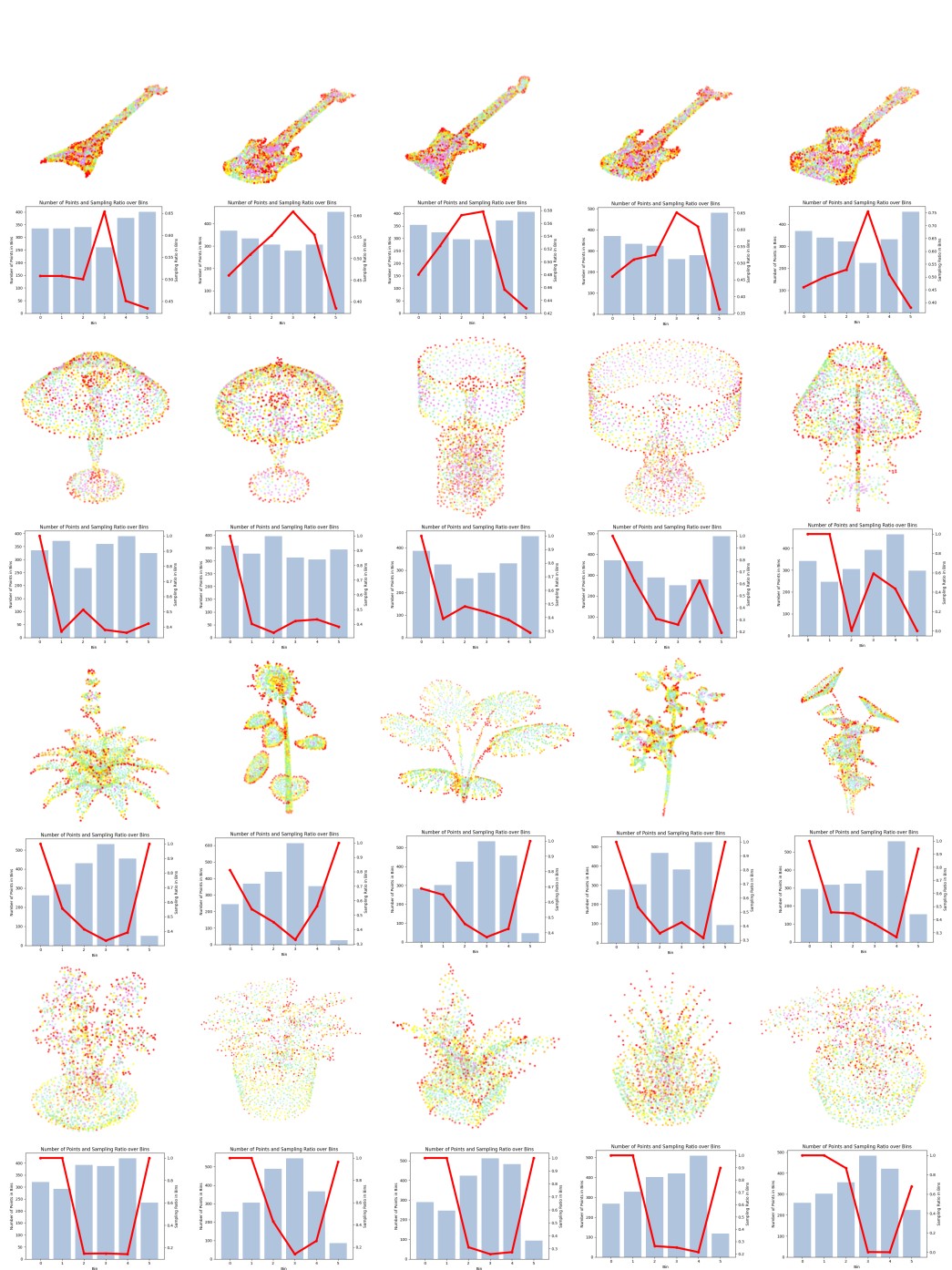

Figure 21: More visualization results of bin partitioning and learned shape-specific sampling strategies. The guitar, lamp, plant, and flower pot categories. Zoom in for optimal visual clarity.

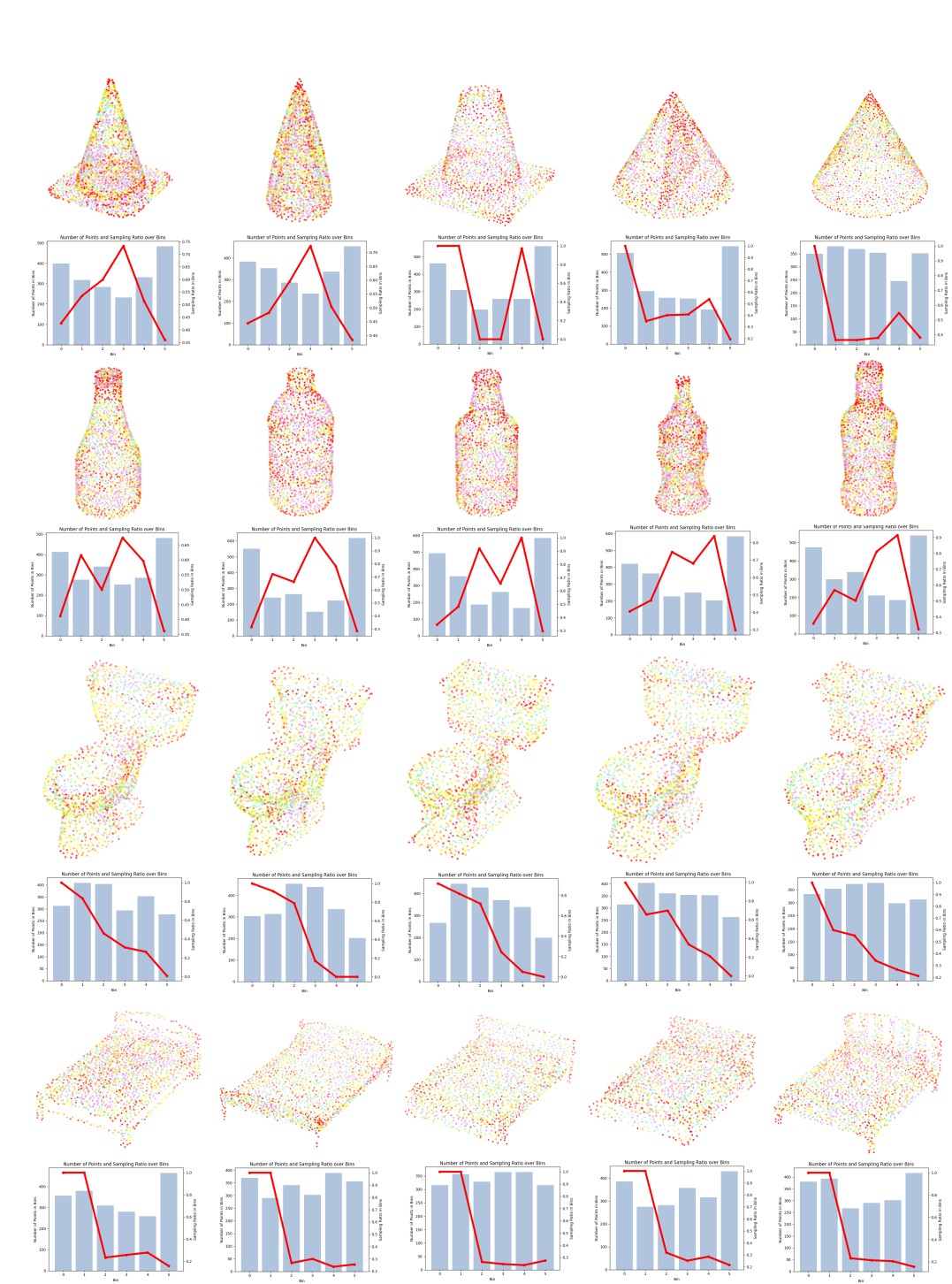

Figure 22: More visualization results of bin partitioning and learned shape-specific sampling strategies. The cone, bottle, toilet, and bed categories. Zoom in for optimal visual clarity.

## J    MORE VISUALIZATION RESULTS OF FEW-POINT SAMPLING

We further provide more visualization results of few-point sampling in Fig. 23 and Fig. 24. No pre-processing with FPS into $2M$ points was performed. From them, we can observe that when sampling very few points from the input directly, APES can only sample points from the sharpest regions in a concentrated manner, while our SAMBLE keeps better global uniformity.

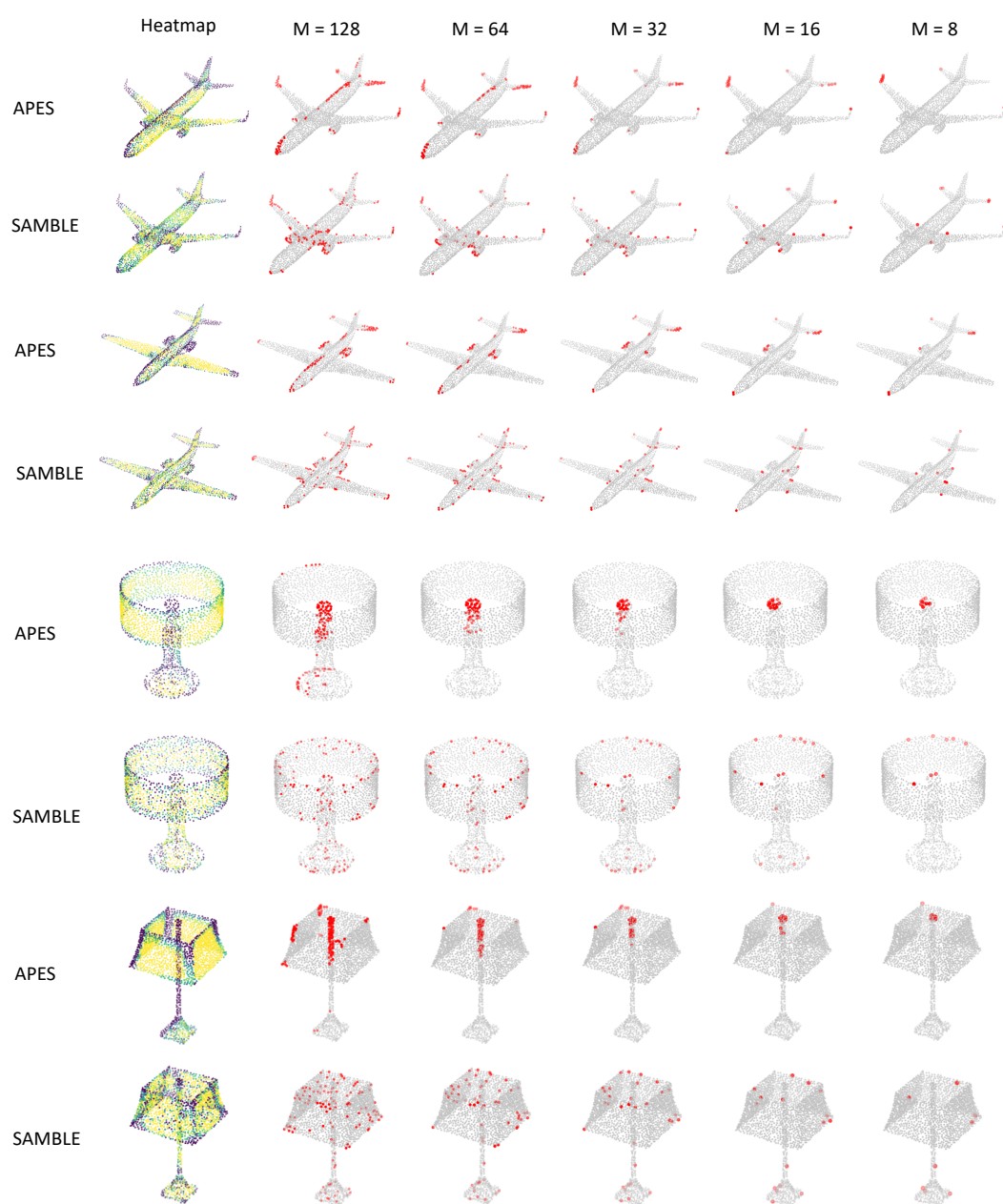

Figure 23: Sampled results of few-point sampling. No pre-processing with FPS into $2M$ points was performed. Zoom in for optimal visual clarity.

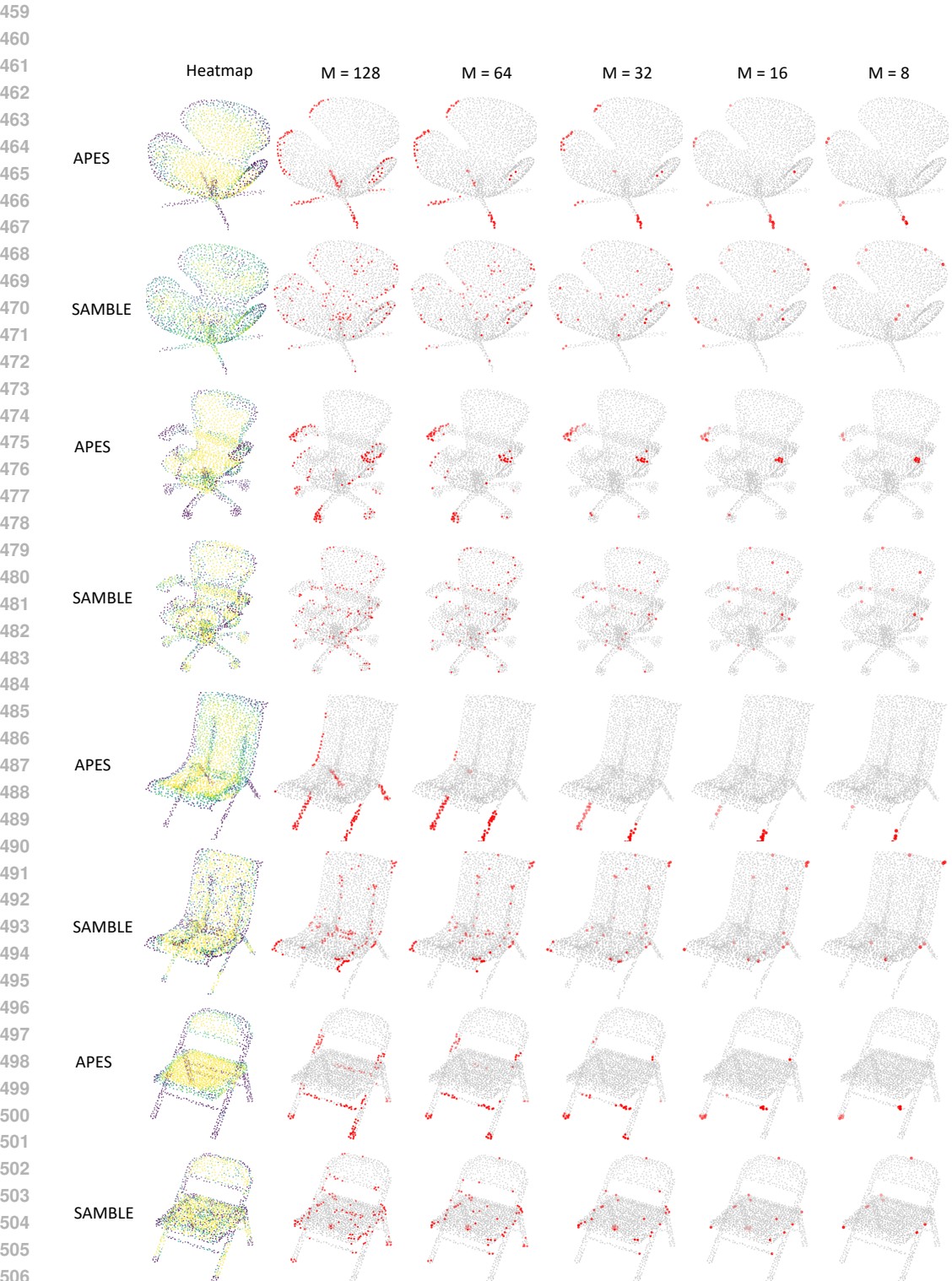

Figure 24: Sampled results of few-point sampling. No pre-processing with FPS into $2M$ points was performed. Zoom in for optimal visual clarity.

