# OpenReview forum: "SAMBLE: Learning Shape-Specific Sampling Strategies for Point Cloud Shapes with Sparse Attention Map and Adaptive Bin Partitioning"
_ICLR.cc/2025/Conference — ICLR 2025 Conference Withdrawn Submission_

### Official Review · Reviewer_iyZW · 2024-10-29

**Soundness:** 2
**Presentation:** 2
**Contribution:** 2
**Rating:** 5
**Confidence:** 3

**Summary:**

In this paper, the authors propose a sparse attention map and bin-based learning method for point cloud sampling. Specifically, sparse attention map is first generated by integrating both local and global information. On top of this attention map, sampling score is calculated and bin-based partition method is proposed. Finally, sampling weights are obtained for each bin for point sampling. Experiments are conducted on different benchmark datasets and the results show that the proposed method is able to achieve shape-specifc sampling of a point cloud. In addition, the proposed method outperforms previous sampling approaches on different tasks.

**Strengths:**

- Superior performance on diverse tasks and datasets.

**Weaknesses:**

- This paper is not easy to follow and some presentation are not very easy to understand. The writting can be further improved.
- In Table 3, why the proposed method produces better accuracy with fewer points while PoinNeXt achieves better performance with more points.
- Several realted works are missing in this paper. These works should be included for discussion and comparison.
[c1] Attention discriminant sampling for point clouds
[c2] Learnable skeleton-aware 3d point cloud sampling
[c3] LTA-PCS: Learnable Task-Agnostic Point Cloud Sampling
[c4] Meta-sampler: Almost-universal yet task-oriented sampling for point clouds
- As the runtime is also critical to a sampling method, it would be better to compare the inference efficiency of the proposed method with previous SOTA ones.
- The sparse attention map is one of the key contribution of this paper. Ablation experiments should be conducted to validate its effectiveness in terms of local and global information, respectively.

**Questions:**

Please see weaknesses.

---

### Official Review · Reviewer_8rCh · 2024-10-31

**Soundness:** 2
**Presentation:** 2
**Contribution:** 3
**Rating:** 5
**Confidence:** 3

**Summary:**

In this work, the authors introduce SAMBLE, a novel point cloud sampling method. SAMBLE begins by constructing a sparse attention map from the point-wise attention data. Sampling scores are then computed from this attention map using an indexing approach, with several indexing strategies tested to determine the optimal one. Finally, the point cloud is partitioned into bins, with sampling conducted within each bin based on the computed scores.

**Strengths:**

1. This method explore different indexing modes for computing sampling scores from the sparse attention map;
2. Learnable vector $v_c$ is introduced to divide the point cloud into different bins, where subsequent sampling will be conducted.
3. Evaluation on the downstream tasks including point cloud classification and segmentation confirms that the proposed method performs better than existing sampling strategies.

**Weaknesses:**

My major concerns about this work is that the presentation is not so clear. Some details are not well defined in this work. For example, is $v_t$ universal for all shapes? In Fig.5, $N * (N+n_b)$ matrix is directly transformed into $N*N$ matrix for score computation. How is this achieved?  I would appreciate it if the authors can provide a clearer explanation about the sampling process in Sec.3.3.

**Questions:**

1. As this framework introduces a attention map to evaluate the scores for points, the complexity of sampling would be $N^2$, which may greatly limit its usage on dense point clouds. Could you present some brief discussions about the sampling efficiency?
 2. This method proposes to sample points by indexing through the  scores calculated from the attention maps. But I cannot quite get how is the corresponding Q, K, V matrices optimized because the indexing process seems to be non-differentiable for the down-stream tasks. Could the authors present some explanations on this?
 3. I am  a little doubtful about the evaluation on the segmentation task. From the results in Fig.7, we can observe that different sampling methods will acquire different sampled points. It is hard to say if it is fair to directly compare the segmentation performances on these different point clouds.
 4. Some related works are not discussed or compared, such as [1], [2];
 [1] REPS: Reconstruction-based Point Cloud Sampling
 [2] Resolution-free point cloud sampling network with data distillation
 5. From Table 3, we can observe that the segmentation performances improve as the number of sampled points reduces, which is somewhat counter-intuitive because the preserved geometrical characteristics should be less. Could the authors briefly explain about this?

**Details Of Ethics Concerns:**

NA.

---

### Official Review · Reviewer_KzoQ · 2024-11-02

**Soundness:** 2
**Presentation:** 2
**Contribution:** 2
**Rating:** 5
**Confidence:** 4

**Summary:**

This paper aims to sample point clouds by shape-specific sampling strategies for striking a balance between the overall shape outline and intricate local details. Specifically, SAMBLE proposes sparse attention map by integrating both local and global information. Besides, SAMBLE adopts a binning strategy and additional learnable tokens to achieve the shape-specific sampling strategies. Finally, SAMBLE achieves superior performance across common point cloud downstream tasks.

**Strengths:**

1. The sampling strategy for point clouds is an important problem.
2. SAMBLE achives superior performance compared with other sampling methods.

**Weaknesses:**

1. The proposed sparse attention map is confusing. Is the sparse attention map equal to a simple combination of local attention and global attention? More experiments are needed to compare the different attention operators (e.g., local attention, global attention, local + global, and sparse attention).
2. I have doubts about the effectivness of the proposed bin-based sampling strategy. As shown in Table 6, there is no noticeable performance difference between setting bin to 1 and 6. Besdies, as shown in Table 2 and 12, the performance of SAMBLE (bin=6) is slightly higher than that of APES, but the computational cost is higher. Therefore, I think the performance improvement may come from the additional computational cost (e.g., local + global attention) rather than the bin-based sampling strategy.
3. I think additional experiments are needed to prove the effectiveness of bin-based sampling strategy. For example, using bin-based sampling strategy on APES.
4. I suggest that authors do not conduct all experiments on only simple synthetic datasets (ModelNet40 and ShapeNetPart). I think more datasets (e.g., S3DIS, ScanNet, KITTI, Waymo) are needed to prove the effectiveness of the proposed sampling method.

**Questions:**

See the weaknesses

---

### Official Review · Reviewer_59De · 2024-11-02

**Soundness:** 2
**Presentation:** 2
**Contribution:** 2
**Rating:** 5
**Confidence:** 4

**Summary:**

This paper introduces SAMBLE: Sparse Attention Map and Bin-based Learning, a method designed to improve point cloud sampling by developing shape-specific sampling strategies. The approach aims to address limitations in current learning-to-sample methods, which often result in unrecognizable sampling patterns or sampling biases that overemphasize shape details. SAMBLE introduces shape adaptability in sampling, taking into account the natural variations in point distribution across different shapes.

**Strengths:**

1. SAMBLE combines sparse attention with bin-based partitioning to offer a shape-specific sampling approach, potentially benefiting applications that require adaptable sampling.
2. The paper includes experiments that demonstrate SAMBLE's effectiveness across several tasks, highlighting its versatility.

**Weaknesses:**

1. Dataset Limitations and Generalizability. The primary concern with SAMBLE lies in its limited evaluation on the ModelNet40 dataset, which consists of high-quality, uniformly distributed point clouds. ModelNet40’s structured nature may not sufficiently challenge sampling algorithms, making it difficult to fully assess SAMBLE’s effectiveness. Additionally, SAMBLE is designed for isolated point cloud shapes, potentially limiting its applicability in complex scenes or datasets where shape boundaries are less defined. Future evaluations on more diverse datasets, such as Objaverse, S3DIS, or KITTI, would better demonstrate its robustness and provide insights into adapting SAMBLE for complex or scene-level point clouds.

2. Efficiency and Scalability. SAMBLE’s bin-based partitioning and adaptive weight computation introduce additional computational overhead, potentially impacting efficiency, especially in large-scale or real-time applications. This complexity may limit SAMBLE’s practicality in scenarios where computational resources are constrained.

3. Lack of 3D-Specific Theoretical Foundation. The theoretical basis in SAMBLE’s introduction relies on 2D image sampling examples with uniformly distributed data, which may not fully represent the complexities of 3D point cloud structures. Unlike 2D images, 3D scenes often have varied and dense distributions across complex structures, making direct analogy insufficient. A more tailored 3D theoretical foundation could enhance SAMBLE's relevance to point cloud sampling and better justify its design choices.

**Questions:**

1. Could the authors provide results on other datasets, like Objaverse, S3DIS, or KITTI, to show SAMBLE’s performance in more complex and real-world scenarios?

2. How does SAMBLE handle noisy or incomplete point cloud data, as often encountered in real applications?

3. How does SAMBLE differ from other attention-based or binning methods?

4. Could the authors discuss how SAMBLE might perform with dynamic or temporally varying point clouds, such as those found in autonomous driving scenarios, and whether any adaptations would be necessary?

5.  Is there a threshold in point cloud density or a minimum number of points below which SAMBLE’s effectiveness significantly drops?

---

### Note · Authors · 2024-11-14

I have read and agree with the venue's withdrawal policy on behalf of myself and my co-authors.